# Understanding Transformer Reasoning Capabilities via Graph Algorithms

**Clayton Sanford**[1,2][*] **Bahare Fatemi**[1]**, Ethan Hall**[3]**, Anton Tsitsulin**[1]**,**
**Mehran Kazemi**[4]**, Jonathan Halcrow**[1]**, Bryan Perozzi**[1]**, and Vahab Mirrokni**[1]
[1]Google Research, [2]Columbia University,[3]Google, [4]Google DeepMind

## Abstract

Which transformer scaling regimes are able to perfectly solve different classes of algorithmic problems? While tremendous empirical advances have been attained by transformer-based neural networks, a theoretical understanding of their algorithmic reasoning capabilities in realistic parameter regimes is lacking. We investigate this question in terms of the network's depth, width, and number of extra tokens for algorithm execution. Our novel representational hierarchy separates 9 algorithmic reasoning problems into classes solvable by transformers in different realistic parameter scaling regimes. We prove that logarithmic depth is necessary and sufficient for tasks like graph connectivity, while single-layer transformers with small embedding dimensions can solve contextual retrieval tasks. We also support our theoretical analysis with ample empirical evidence using the GraphQA benchmark. These results show that transformers excel at many graph reasoning tasks, even outperforming specialized graph neural networks.

## 1 Introduction

The transformer neural network architecture, which was initially introduced for neural machine translation [6, 77], quickly became the standard neural network architecture across many fields, powering recent breakthroughs in language modeling [20, 66, 12, 76, 3, 75, 67], computer vision [21, 49], and natural sciences [36, 53]. Across fields, transformers have superseded other architectures, surpassing them in downstream performance while maintaining reasonable computational footprint.

How can we analyze reasoning capabilities of neural networks? One approach is to study algorithms executable with their internal representations. Neural algorithmic reasoning [30, 89, 34, 37, 78] is a field of research dedicated to exploring such capabilities. Algorithmic execution is desirable because models use it to generalize out-of-distribution [92, 46] and scale to larger problem sizes [93].

In this work, we focus on classes of transformers solving algorithmic reasoning problems on graphs. Why graph problems in particular? Recent research by Besta et al. [10] suggests that graphs are an ideal abstraction of complex reasoning with dependencies, including chain-of-thought [82] and its generalizations [86, 17, 41, 40]. Furthermore, we investigate graph reasoning in order to evaluate transformer capabilities compared to specialized models that explicitly capture the structure of data. Graph neural networks (GNNs) [72, 44, 28, 7, 15] offer strong baselines for algorithmic reasoning [79, 13] and comprehensive theoretical frameworks for their capabilities and limitations [51, 84]. The complexity of graph reasoning tasks varies greatly; some are easily solved by non-graph architectures and others require sophistication beyond standard GNNs [90]. This makes graph tasks a compelling testbed for both theoretical and empirical evaluations.

While transformer-based models are commonly optimized for downstream performance [38, 32, 64], theoretical investigations of transformer capabilities in realistic parameter regimes have been limited.

---

[*]Correspondence: `chsanford@google.com` or `baharef@google.com`.

38th Conference on Neural Information Processing Systems (NeurIPS 2024).

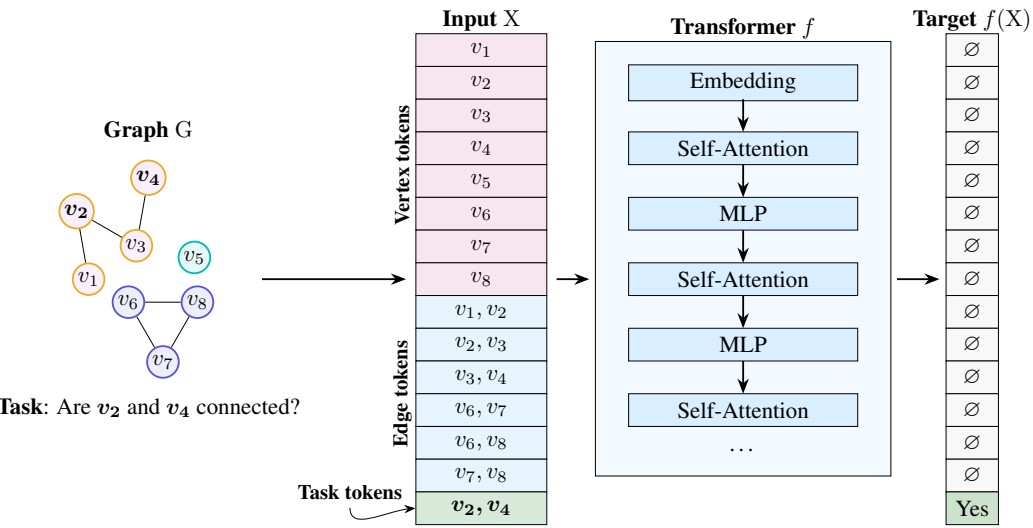

Figure 1: The graph encoding scheme employed in our theoretical and empirical analysis that presents a graph reasoning task (e.g. connectivity) as a tokenized input to a standard transformer model.

We analyze the capabilities of standard transformer models to solve graph reasoning tasks by employing a graph tokenization scheme similar to [42]; see Figure 1 for a graph encoding example. We generalize the insights of [71]—which established that graph connectivity can be computed in a more depth-efficient manner with transformers than GNNs—to broader families of graph reasoning tasks. We also study the representational impacts of *blank tokens*, which can be thought of as a theoretically-tractable version of either chain-of-thought prompting [82], scratch space [61], or pause/filler tokens [29, 65].

We conduct an extensive study of graph reasoning problems with transformers from both theoretical and empirical perspectives. We summarize our principal contributions as follows:

1. We introduce a novel *representational hierarchy* of graph reasoning tasks that formalizes reasoning capabilities of transformers in several realistic parameter scaling regimes. This includes two graph reasoning task classes—which we refer to as the *parallelizable* and *search tasks* and include well-known algorithmic tasks like *connectivity* and *shortest path* respectively.
2. We prove that logarithmic-depth transformers are necessary and sufficient to solve parallelizable tasks in a highly parameter-efficient manner; similar constructions for search tasks employ much larger networks. We distinguish both of these classes from an easier family of problems—*retrieval tasks*, such as *node count* and *edge existence*—by showing that retrieval tasks can be efficiently computed by single-layer transformers, while parallelizable and search tasks cannot.
3. We empirically validate our representational contrast by showing that transformers outperform GNNs on tasks that require the analysis of long-range dependencies, including parallelizable tasks like connectivity. Our experiments suggest that GNNs perform better on simpler tasks that require only local analysis of neighboring nodes in low sample-complexity regimes, benefiting from their well-aligned inductive bias.

## 2   Related work

**Fundamental capabilities of transformers.**   Early representational research [88, 62, 81] established the universality of transformers by simulating Turing machines step-by-step, albeit in an unrealistic polynomial-depth scaling regime. These universality results were extended to bounded-depth transformers with chain-of-thought tokens [56, 52], but with a persistent gap between positive and negative results. A more fine-grained theoretical approach established the limitations of bounded-size transformers by relating them to threshold circuits, implying that L-complete tasks like graph connectivity are unsolvable by constant-depth transformers [55, 54, 57]. However, these circuit complexity reductions are not bidirectional and their positive results pertaining to classes of regular languages [31] reveal little about when tasks like connectivity *can* be expressed.

The closest analytical framework to ours characterizes transformers as distributed computing protocols by quantifying the informational bandwidth of their self-attention units. This line of work sharply separates transformers from other architectures and provides width and depth separations [70, 71]. The ability of transformers to simulate finite-state automata in logarithmic depth provides a further roadmap for how transformers efficiently leverage parallel computation [48].

Recent work [18] similarly focuses on understanding the capabilities of transformers to solve graph algorithms, but they focus instead on *looped* transformers, where a constant-size transformer model is iterated up to a polynomial number of times in the size of the input graph. While their constructions for tasks like graph connectivity are more parameter-efficient than our own, this comes at the cost of a very high depth, which presents both run-time and learnability concerns. Our depth-efficient constructions emerge by simulating parallel graph algorithms, rather than well-known serial algorithms, such as Dijkstra's.

**Empirical analysis of transformer reasoning capabilities.** Multiple empirical investigations explore the algorithmic capabilities of transformer-based models [91, 47, 46]. Graph reasoning problems have been used to evaluate capabilities of transformer-based LLMs [24, 80, 87, 74, 33] including works with graph-specific GNN adapters [14, 63]. In the empirical part of our study, we use the GraphQA dataset [24]—which was evaluated initially on LLM prompting appraoches and later on GNN-augmented prompts [62]—to validate our complexity hierarchy and compare transformers with GNNs and prompt-based LLMs on a selection of graph reasoning tasks.

**Transformers and graph neural networks.** GNNs provide a useful benchmark for model expressivity on graph inputs. For instance, the inability of GNNs to efficiently solve "global structure" tasks like subgraph connectivity made evident by a connection to the CONGEST distributed computing model [51]. A further connection [59] to the Weisfeiler-Lehman (WL) isomorphism heuristic [83] captures the inability of feature-less GNNs to distinguish certain graph instances [84]. The feature-less assumption is burdensome and representationally costly; GNNs with randomly intialized node features are known to be universal [2]. See Appendix D for a further discussion of the representational limitations of GNNs.

Graph transformers [22, 68, 58] integrate the transformer architecture with graph-structured data. While GNNs can simulate the WL test [1], transformers can simulate GNNs and exceed their WL limitations [90]; they are strictly more expressive than 2-WL [42]. Despite our focus on standard transformers, our empirical results in Section 4 include comparisons to a wide range of GNNs.

## 3 Hardness taxonomy of transformer graph reasoning tasks

We provide our core result: a rigorous quantification of the hardness of graph reasoning tasks for transformer-based models. While graph reasoning tasks, like other algorithmic problems, can be categorized into well-known computational and circuit complexity classes (e.g. $\mathsf{TC}^0$, L, NL, NP, NP), the relationship between membership in these classes and the hardness of solving a task with a parameter-efficient neural network is not immediately obvious. Our hierarchy bridges the gap between these worst-case computational classes and the representational capabilities of bounded-size transformers of different parameter scaling regimes. These regimes include transformers whose depth $L$ scales with the input sequence length $N$; this contrasts with most theoretical results, which study on the constant-depth regime.

Our positive and negative theoretical results employ the transformer model of [71], which is presented in Appendix A.1 and assumes that the embedding dimension $m$ grows less rapidly than than the sequence length $N$ and that the multi-layer perceptrons (MLPs) are arbitrary functions. The result is a model of a transformer as a *bounded-capacity communication protocol*. In this model, arbitrary functions of each individual embedding vector can be computed, but the interactions between these vectors are restricted by the low-rank of the attention matrix. The relevance of this model is motivated by the rapid scaling in the context lengths of modern transformers in recent years and the high ratio of MLP parameter count to the embedding dimension each operates on.

This model also permits the inclusion of blank *pause token* inputs of [29], which provide additional computational power to the transformer model by extending the computational "tape" without introducing new information about the input.

(a) The complexity hierarchy

| Task class | Example tasks | Complexity |
|---|---|---|
| Retrieval (§3.3) | Node count | D1 |
| $L = 1$ | Edge count | D1 |
| $m = O(\log N)$ | Edge existence | D1 |
| | Node degree | D1 |
| Parallelizable (§3.1) | Connectivity | LD |
| $L = O(\log N)$ | Cycle check | LDP∩ LDW |
| $m = O(N^\epsilon)$ | Bipartiteness | LDP∩ LDW |
| Search (§3.2) | Shortest path | LDW |
| $L = O(\log N)$ | Diameter | LDW |
| $m = O(N^{1/2+\epsilon})$ | | |

(b) Example tasks and their complexity.

Figure 2: A summary of the theoretical hierarchy of Section 3 that visualizes which type of graph reasoning tasks can be solved in which transformer scaling regime (Depth1 (D1), LogDepth (LD), LogDepthWide (LDW) and LogDepthPause (LDP)).

These results divide the tasks that are we empirically investigate in Section 4 into three difficulty families based on the hardness of solving the task with parallel computation.

1. **Retrieval tasks**—including node count, edge count, edge existence, and node degree—are tasks that can intuitively be solved by a single lookup step or global aggregation. These are the easiest tasks in our framework, and we show in Section 3.3 that these retrieval tasks can be computed by a single-layer transformer with small embedding dimension. (In contrast, all other examined tasks cannot be solved in that regime.)
2. **Parallelizable tasks**—including connectivity, connected nodes, and cycle check from the experimental results and a host of other graph reasoning tasks, such as bipartiteness, planarity, and minimum spanning forest—are non-trivial tasks that can be solved efficiently in a parallel computation setting. Section 3.1 establishes that these tasks can be solved by bounded-size transformers with logarithmic depth.
3. **Search tasks**—including shortest path and other tasks like diameter and directed reachability—comprise a harder family of tasks that are less easily solved by parallel algorithms. In Section 3.2, we prove that these tasks belong in an equivalence class and exhibit large-model scaling regimes where they can be computed.

The representational hardness of these classes is quantified by several results that determine whether transformers that obey different parameter scaling rules can compute them. We define the following scaling regimes for the depth $L$, embedding dimension $m$, and number of "pause" tokens $N'$ of a family of transformers as a function of the size of the input graph, $N = O(|V| + |E|)$.

- Depth1 (D1): Single-layer multi-headed transformers with small embedding dimension $m = O(\log N)$ without any pause tokens.

- LogDepth (LD): Transformers with depth $L = O(\log N)$, embedding dimension $m = O(N^\epsilon)$ for any fixed $\epsilon > 0$, and no pause tokens.

- LogDepthPause (LDP): Transformers with the same depth and width constraints as LogDepth, with at most $N' = \text{poly}(N)$ blank "pause" tokens appended to the input sequence.

- LogDepthWide (LDW): Transformers with depth $L = O(\log N)$, embedding dimension $m = O(N^{1/2+\epsilon})$, and no pause tokens.

Our positive and negative results that relate scaling regimes and graph reasoning tasks are displayed in Figure 2. We present high-level result summaries in the following sections with proofs in Appendix.

The most technically significant results concern LogDepth models and are provided in Appendix B. These bounds are consequences of Theorem 1, an improved analysis of the relationship between

transformers and the massively parallel computation (MPC) distributed computing model of [39][2]. This connection between MPC and transformers is a sharp improvement of a similar result by [71].

**Theorem 1** (Simplified version of Theorem 8). *For constant $\delta, \epsilon > 0$, any $R$-round MPC protocol with $N$ machines with $O(N^\delta)$ bits of local memory each can be simulated by a transformer of depth $L = O(R)$ and embedding dimension $m = O(N^{\delta+\epsilon})$.*

Results pertaining to Depth1 are stated in detail and proved in Appendix C. In addition, we discuss the triangle counting task (and the more general clique counting task) in Section 3.4, where we show a distinct result for much smaller depth ($L = O(\log \log N)$) that includes pause tokens.

### 3.1 Parallelizable tasks and LogDepth transformers

We define the family of *parallelizable tasks* to consist of graph reasoning tasks that are L-complete and are equivalent to graph connectivity in $O(1)$-rounds, as proved by [60]. This family includes (but is not limited to): graph connectivity, minimum spanning forest, cycle detection, $st$-connectivity, number of connected components, graph bipartiteness, planarity testing, and one-cycle vs two-cycles testing. While graph connectivity and minimum spanning forest were shown to be computable by logarithmic depth transformers with small polynomial width (LogDepth) by previous work [71], this poses broader questions: Are all parallelizable graph reasoning tasks computable by a transformer of logarithmic depth and small embedding dimension? And do sub-logarithmic-depth transformers exist that solve any parallelizable tasks?

We first show that all parallelizable tasks can be solved in two logarithmic-depth scaling settings.

**Theorem 2.** *For any parallelizable task, there exists transformers in* LogDepthPause *and* LogDepthPause *that solve the task.*

Theorem 2 is stated formally as Theorem 18 and proved in Appendix B.2.1. Both components of the theorem are a consequence of a novel relationship between the MPC model and transformers (Theorem 1) and the analysis of MPC protocols for graph reasoning tasks by [60]. The LogDepthPause result is the direct implication of an $O(1)$-round MPC equivalence (Theorem 9) between all parallelizable tasks and an MPC protocol that solves the connectivity task (Theorem 11). The LogDepthWide bound is a consequence of Theorem 15, which shows that all languages in $NC^2$ (including all languages in L and NL) can be evaluated by an MPC protocol with $O(\log N)$ rounds and with local memory $O(N^{1/2+\epsilon})$ [60].

We further prove the conditional optimality of logarithmic depth.

**Theorem 3.** *Conditional on Conjecture 13, any transformer that solves some parallelizable task with width $mH = O(N^\epsilon)$ and pause tokens $N' = \text{poly}(N)$ must have depth $L = \Omega(\log N)$.*

This result (stated formally as Theorem 19 is a combination of the conditional depth lower bound on graph connectivity by [71] and the $O(1)$-round equivalence between all parallel tasks.

### 3.2 Search tasks

*Search tasks* are similarly defined to be those that are NL-complete and equivalent to shortest path in $O(1)$-rounds of MPC and include shortest path, strong connectivity, $st$-reachability, radius, diameter, and median. Like before, the $O(1)$-round MPC equivalence translates to an $O(1)$-depth equivalence in transformers. We give a similar positive result for LogDepthWide transformers; whether these tasks can be solved by LogDepthPause transformers is unknown.

**Theorem 4.** *For any search task, there exists a transformer in* LogDepthWide *that sovles the task.*

This theorem, which is restated in Appendix B.2.2 as Theorem 21, is also an immediate consequence of Theorem 15.

While the minimum depth of a transformer with small embedding dimension that solves a search task is not identified, we prove that the minimum depth needed to solve some all search task is approximately equivalent in Theorem 22.

---

[2]MPC is a theoretical model of the MapReduce computational paradigm that distributed computation among a large number of machines with restricted local memory size and communication bandwidth. We formally introduce the MPC model in Appendix B.1.

### 3.3 Retrieval tasks and `Depth1` transformers

Graph tasks whose algorithms consist of a single look-up or aggregation step can be efficiently solved by single-layer transformers. This result assumes that the graph $G = (V, E)$ is encoded as some input sequence $X$ of length $N = O(|V| + |E|)$ that expresses each edge and vertex a single token. This encoding scheme is detailed in Appendix C.

**Theorem 5.** *For any retrieval task (including node count, edge count, edge existence, and node degree) there exists a transformer in* `Depth1` *that solves the task.*

We formalize this statement in Theorem 36 and prove it their in Appendix C.1. These rely on proving the existence of a useful input MLP $\phi$ that precomputes embeddings with useful structure for all retrieval tasks.

In contrast, we show that a collection of parallelizable and search tasks cannot be efficiently solved by transformers in `Depth1`.

**Theorem 6.** *Any single-layer transformer that solves the graph connectivity, shortest path, or cycle detection task has width satisfying* $mH = \tilde{\Omega}(N)$.

The proof of the formal counterpart of this statement (Theorem 38) appears in Appendix C.2 and is a consequence of a standard communication complexity argument. A more generalized result than Theorem 6 was proved by [57], which establishes that all problems outside of $\text{NC}^1$—which include all L-complete and NL-complete languages, and hence, all parallelizable and search tasks—cannot be solved by *constant-depth* transformers with polynomial width because they cannot be computed by $\text{TC}^0$ (constant-depth threshold circuits). We nonetheless include this theorem to demonstrate a clean lower bound that applies to very simple input graph instances.

### 3.4 Triangle counting

We finally construct depth-efficient transformers for triangle counting due to the MPC algorithms of [11]. Unlike previous positive results, which applied uniformly across all graphs instances of bounded size, the complexity of the corresponding transformers for triangle counting is a function of the arboricity[3] of the input graph. When the arboricity grows sub-polynomially with $N$—as is the case for bounded-degree graphs—no pause tokens are necessary. Unlike the parallelizable and search classes of problems, strictly sub-logarithmic depth is attainable with pause tokens, even for worst-case graphs.

**Theorem 7.** *There exists a transformer that computes the number of triangles in any input graph of arboricity $\alpha$ and has embedding dimension $m = O(N^\epsilon)$, depth $L = O(\log \log N)$, and pause tokens*

$$N' = \begin{cases} O(\alpha N^{1-\epsilon}) & \text{if } \alpha = \Omega(N^\epsilon) \\ 0 & \text{otherwise.} \end{cases}$$

This result is a special case of Theorem 23, a more general result about clique counting that appears in Appendix B.2.3.

**Theoretical conclusions.** These results provide a tight characterization of the the reasoning capabilities of transformers whose depth, width, and input padding conform to different scaling regimes. They strengthen the established connection between transformers and massively parallel computation (MPC) [71] and generalize the resulting representational bounds to broader categories of graph tasks. We conclude that the logarithmic-depth regime is apt for for considering tasks in L and NL, which had previous illuminated the limitations of transformers with constant depth and a limited number of chain-of-thought tokens [56]. While expressivity does not imply learnability, these theoretical benchmarks sharply characterize the fundamental limitations of transformers and coincide with experimental results conveyed in the subsequent section.

Our theoretical model of a transformer has certain limitations, namely the universality of the multi-layer perceptron units. When maximally exploited (e.g. by solving NP-hard problems within each MLP), this assumption exaggerates the computational capabilities of transformers. However, the

---

[3]The *arboricity* is the minimum number of spanning forests needed to cover the graph, which grows at most linearly with the degree of the graph.

local computational complexity of the MPC algorithms employed in this paper tend to be strongly sublinear in the size of the graph input (see, e.g. the connectivity algorithms of Andoni et al. [4]), which implies the existence of compact ReLU circuits for each of the model's MLPs. Furthermore, the ratio of model parameters that belong to MLPs in state-of-the-art language modelsis known to be high, which suggests that it is reasonable to assume that MLP units in our theoretical model are algorithmically rich. We would be interested in pursuing further theoretical research that combine the computational model [54] and the communication model of [71] to assess the capabilities of transformers whose MLPs are represented as bounded-size boolean circuits.

# 4 Empirical graph reasoning capabilities

We further illuminate the reasoning capabilities of transformers by conducting an empirical investigation of the abilities of a variety of neural architecture and training settings to learn graph algorithmic tasks. We use the GraphQA benchmark tasks [24] for our experiments. We evaluate standard autoregressive transformers—both small models (at most 60M parameters) trained from scratch and fine-tuned (FT) T5-11B model (with 11B parameters) [66]. For the fine-tuned models, we explore task-specific fine-tuning—and contrast those results with graph neural networks (GNNs) and prompting-based methods on pre-trained LLMs.

These experimental results validate key tenets of our theoretical results and demonstrate the utility of transformers' algorithmic reasoning capabilities. Our principal empirical conclusions are as follows:

1. **Transformers excel at global reasoning tasks.** Transformers outperform GNNs on tasks that require efficiently aggregating information about distant nodes in a graph, such as connectivity and shortest path.

2. **GNNs uncover local graph structure with few samples.** While transformers are capable of efficiently expressing all graph learning tasks under investigating, the structural limitations of GNNs provide them with favorable inductive biases for intrinsically local tasks, such as cycle check and node degree, and permit them to outperform transformers in a low-sample regime.

3. **Trained transformers outperform LLM prompting.** Transformers trained to explicitly solve graph reasoning tasks consistently attain greater accuracy across tasks than a variety of prompting strategies applied to more recent larger LMs.

A comprehensive evaluation of each GraphQA task on every training setting appears in Appendix E, in addition details about transformer training, the GraphQA benchmark, and alternative GNN and prompting approaches.

## 4.1 Transformers excel at global reasoning tasks

As indicated in Section 3, graph reasoning algorithms can be categorized based on the extent to which they entail aggregating "local" information about nodes and their immediate neighbors or modeling "global" connections between nodes separated by a long distances. This section investigates the following question about transformers and long-distance reasoning:

*When do transformers outperform GNNs on tasks that require global reasoning?*

We consider two tasks that require reasoning across long distances in a graph instance: evaluating **connectivity** and computing the **shortest path** between a pair of nodes. Neither of these tasks can be solved by only investigating the neighbors of the source and sink node, which therefore implies that some analysis of global graph structure is necessary.

Figure 3a displays the accuracy of a variety of trained transformers and GNNs on the connectivity task contrastsing the performance of all such models when trained on 1,000 and 100,000 graph connectivity instances. In the most restricted sample complexity regime, trained GNNs are consistently more accurate than the small transformer; however, increasing the number of training samples yields a far more substantial improvement in the performance of the small transformer, which outperforms all GNNs trained on 100,000 samples. Notably, the pre-trained transformer, fine-tuned on just 1000 training instances, nearly solves the connectivity task. This suggests significant enhancements due to the larger model size and the data-rich fine-tuning phase. Figure 3b plots the training and test error of ten small transformers trained on connectivity datasets of increasing size and reveals a sharp and

|  | # of training samples | |
|---|---|---|
| **Model** | **1K** | **100K** |
| GCN [44] | 83.8 | 83.8 |
| MPNN [28] | 94.0 | 94.4 |
| GIN [84] | 93.8 | 94.0 |
| 60M transformer | 92.9 | 98.0 |
| 11B transformer (FT) | **98.4** | — |

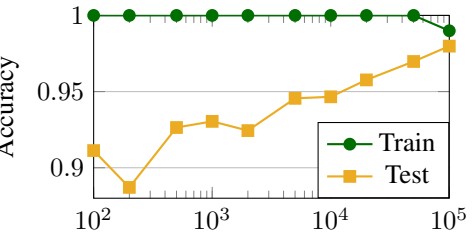

(a) Connectivity classification accuracy for trained transformers and GNNs.

(b) Connectivity classification accuracy of 60M transformers by training set size.

Figure 3: Accuracy of a variety of trained transformers and GNNs on the connectivity task.

continual improvement in accuracy. The fine-tuned T5 transformer has similar accuracy to the most sample-rich small transformer and exceeds that of all GNNs.

On the other hand, Table 1 demonstrates that the MPNN GNN models outperform small transformers when trained to compute the shortest path, even on larger training datasets. However, the fine-tuned T5 model has far higher accuracy than all alternatives, even when trained only 1000 samples.

**Theoretical interpretation:** Because connectivity is the prototypical example of a task in the parallelizable class and can thus be efficiently implemented by LogDepth transformers with very small width (Theorem 2), the fact that small transformers succeed in solving nearly all connectivity instances is illuminating but not surprising. In contrast, message-passing GNNs are unable to solve connectivity in a similarly depth- and width-efficient manner due to fundamental capacity limitations.[4]

|  | # of training samples | |
|---|---|---|
| **Model** | **1K** | **100K** |
| GCN [44] | 50.2 | 55.0 |
| MPNN [28] | 66.8 | 72.6 |
| GIN [84] | 54.0 | 58.6 |
| 60M transformer | 57.4 | **97.1** |
| 11B transformer (FT) | 92.8 | — |

Table 1: Transformers vs GNNs on shortest path: Fine-tuned large transformers outperform other transformers and GNNs, even the alternatives are trained on much larger training sets.

Shortest path belongs to the search class of graph reasoning tasks and is NL-complete. Theorem 4 implies that shortest path is computable by LogDepthWide transformers, which are likely to require very large embedding dimension to be learnable by finite samples. This task can only be computed in a depth- and parameter-efficient manner if a variety of search tasks, including all-pairs shortest-path and diameter, are as well (Theorem 22). The fact that only the pre-trained model has nearly optimal performance on shortest path reinforces the theoretical intuition that solving shortest path requires a very large number of model parameters.

## 4.2 GNNs uncover local graph structure with few samples

While small transformers outperform GNNs on graph reasoning algorithms that entail analysis of long-range graph structure, their empirical successes are not uniform. Here, we investigate the following question:

*When do GNNs outperform transformers on graph reasoning tasks?*

Figure 3b and Table 1 demonstrate that GNNs outperform small transformers in the low-sample regime, despite the sufficient expressivity of transformers. This gap in performance, which is reinforced for the **node degree** and **cycle check** tasks in Table 2, suggests that GNNs have a beneficial

---

[4]See Appendix A.2 for a discussion of prior work on theoretical relationships between GNNs and both the Weisfeiler-Leman graph isomorphism test [84] and the CONGEST distributed computing model and their implication that GNNs cannot solve connectivity in a parameter-efficient manner.

| | Method | Retrieval tasks | | | | Parallelizable Tasks | | Search Tasks | Subgraph Counting |
|---|---|---|---|---|---|---|---|---|---|
| | | Node count | Edge count | Edge existence | Node degree | Connectivity | Cycle check | Shortest path | Triangle counting |
| Prompting | ZERO-SHOT [24] | 21.7 | 12.4 | 44.5 | 14.0 | 84.9 | 76.0 | 11.5 | 1.5 |
| | ZERO-COT [24] | 14.6 | 9.4 | 33.5 | 10.4 | 73.5 | 32.3 | 33.6 | 12.7 |
| | FEW-SHOT [24] | 25.3 | 12.0 | 36.8 | 17.4 | 79.4 | 37.4 | 22.7 | 3.0 |
| | COT [24] | 27.6 | 12.8 | 42.8 | 29.2 | 45.2 | 58.0 | 38.6 | 8.1 |
| | COT-BAG [24] | 26.9 | 12.5 | 37.3 | 28.0 | 45.2 | 52.1 | 40.4 | 8.1 |
| Ours | 60M transformer-1K | 100.0 | 100.0 | 67.6 | 31.5 | 92.9 | 97.1 | 57.4 | 33.4 |
| | 60M transformer-100K | 100.0 | 100.0 | 96.1 | 91.7 | 98.0 | 98.0 | 97.2 | 40.5 |
| | 11B transformer (FT)-1K | 100.0 | 45.0 | 100.0 | 68.8 | 98.4 | 98.0 | 92.8 | 26.0 |

Table 3: Comparison of GraphQA task accuracies of transformers explicitly trained for graph reasoning and LLMs with a variety of prompting strategies.

*inductive bias* for learning graph reasoning tasks that can be solved by attending exclusively to local heuristics.

Just as the bounded kernel size of convolutional neural networks (CNNs) enables the sample-efficient learning of relevant textures and gradients for image analysis, message-passing GNNs are unable to send messages instantaneously across multiple edges, which simplifies a search of the space of "one-hop" graph algorithms and represents a positive inductive bias. In contrast, the ability of transformers to send information between any pair of input tokens—and the alternative inductive biases suggested by the input positional encoding—likely induces a steeper sample complexity to learn node degree.

| | Node Degree | | Cycle Check | |
|---|---|---|---|---|
| Model | 1K | 100K | 1K | 100K |
| GCN [44] | 9.8 | 9.4 | 83.2 | 83.2 |
| MPNN [28] | 99.4 | **99.8** | 99.0 | **100.0** |
| GIN [84] | 36.2 | 37.8 | 98.8 | 83.2 |
| 60M transformer | 31.6 | 91.7 | 97.1 | 98.0 |
| 11B transformer (FT) | 68.8 | — | 98.0 | — |

Table 2: Transformers vs GNNs on cycle check and node degree: GNNs are favorably biased for local structure.

**Theoretical interpretation:** While model expressivity is necessary for learnability, it is not sufficient. The locality constraints of message-passing GNNs likely provides a favorable inductive bias for learning tasks like node degree with an exclusively on local structure that makes learning these tasks possible in a sample-efficient manner. While cycle check is more representationally difficult for GNNs than transformers in the worst case (see Appendix A.2), the random graphs sampled for the GraphQA benchmark have very small cycles (Figure 7) and do not resemble the large-diameter worst-case instance.

### 4.3 Trained transformers outperform LLM prompting

Large language models (LLMs) are regularly evaluated by their reasoning abilities, and it remains an open research question to determine what kinds of training data best teaches models to solve logical problems. We investigate the extent to which LLMs can already perform graph reasoning tasks without being trained explicitly to do so.

> *Do transformers trained explicitly to solve graph reasoning tasks outperform prompt-tuning approaches on much larger LLMs?*

In Table 3, we contrast the capabilities of trained transformer models with several prompt-based approaches to querying LLMs. Task-specific transformers—including the fine-tuned 11B transformers—consistently dominated the prompt-based approaches, despite the vast difference in parameter count and the almost certain presence of graph reasoning in the LLM's corpus.

**Theoretical interpretation:** While the representational capabilities of LLMs to solve reasoning tasks is much greater than that of small transformers, this performance gap suggests that their effective reasoning capacity is much weaker and that it may be improved by a richer training corpus that includes synthetic tasks.

Finally, we observe that the near-perfect performance of trained transformers on the node count, edge count, and edge existence is consistent with the representational easy of those tasks, as suggested by the existence of efficient Depth1 transformer implementations.

# 5 Conclusion

This paper provides a comprehensive evaluation of transformer models' graph reasoning capabilities, shedding light on their effectiveness across diverse graph reasoning tasks. By introducing a novel representational hierarchy, the study distinguishes between retrieval, parallelizable, and search reasoning tasks and offers insights into the performance of transformers at varying levels of granularity. The empirical investigation reveals that transformers exhibit strong performance in graph-based reasoning problems, often matching or surpassing specialized graph models. Furthermore, the study highlights transformers' exceptional ability to capture global graph patterns effectively, showcasing their capability in understanding long-range dependencies, a critical factor in solving tasks involving global graph structures. Overall, this work crystallizes precise representational trade-offs that reflect the fundamental reasoning capabilities of transformers and demonstrates that the tasks used to quantify those capabilities are indeed learnable in a sample-efficient and parameter-efficient manner.

While the hierarchy introduced by this work effectively separates graph algorithmic tasks into distinct equivalence classes with significant implications for their computability by transformers, several questions remain for future research. We focused on graph reasoning tasks due to their relevance to the broader context of transformers, GNNs, and parallel algorithms. However, the complexity classes presented here could potentially be extended to a wider range of algorithmic problems. While our assumption of unbounded-size MLPs provides strong lower bounds, further research into whether parallelizable tasks can be represented by transformers with bounded-size MLP units would complement this existing work. Broader experimental results that empirically evaluate the scaling laws would more directly assess the relevance of representational theoretical results to learnability.

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

# A  Theoretical preliminaries

For some vector $v \in \mathbb{R}^N$, let

$$\text{softmax}(v) = \frac{1}{\sum_{i=1}^N \exp(v_i)} \left(\exp(v_1), \ldots, \exp(v_N)\right) \in \mathbb{R}^N.$$

Let $e_i \in \mathbb{R}^m$ denote the $i$th elementary vector, $\vec{1} = (1, \ldots, 1) \in \mathbb{R}^m$ and $\vec{0} = (0, \ldots, 0) \in \mathbb{R}^m$. Let $[n] = \{1, \ldots, n\}$.

## A.1 Transformer models

We use a similar theoretical definition of a multi-layer transformer model to the one employed by [71]. We use a *bounded communication* theoretical model of the standard bidirectional transformer of [77], which assumes that the principal limitation of the transformer's representational power is the low rank of the self-attention matrix. Several theoretical assumptions are necessary to model transformers in this regime:

- the interleaved element-wise multi-layer perceptron (MLP) units compute arbitrary functions;
- the embedding dimension $m$ and number of heads per layer $H$ are much smaller than the sequence length $N$; and
- all model parameters, inputs, and intermediate products can be represented with $O(\log N)$-bit numbers.

These assumptions are justified in greater detail by [71].

Furthermore, we also model transformers as having $N'$ additional "pause token" inputs, which are embeddings that contain no information relevant information about the input instance, but which provide a longer "work tape" that can be utilized for computational purposes [see e.g. 29, 65]. While structurally similar to chain-of-thought reasoning [82], these pause tokens are not determined by multiple auto-regressive passes over the transformer and do not encode useful information as input. Such pause tokens exist both in the theoretical results of Section 3 (as blank tokens at the end of the sequence) and in the empirical results of Section 4 (as placeholders between the graph representation and the task query for graphs that can be represented in fewer tokens than the maximum sequence length).

**Definition 1.** Let $\mathsf{Transformer}_{m,H,L}^{N,N'}$ denote the family of all bidirectional *transformers* with embedding dimension $m$, number of heads $H$, and depth $L$, which operate on input sequences on length $N$ with $N'$ blank pause tokens appended to the end[5]. Any transformer $f \in \mathsf{Transformer}_{m,H,L}^{N,N'}$ is a function of the form $f : \mathbb{R}^{N \times d} \to \mathbb{R}^N$ for some input dimension $d$, which parameterized by query, key, and value matrices

$$Q_{\ell,h}, K_{\ell,h}, V_{\ell,h} \in \mathbb{R}^{m \times m}, \text{ for } \ell \in [L], \ h \in [H],$$

and arbitrary element-wise MLPs with positional embeddings $\phi^0, \ldots, \phi^L$ and $\psi$ with

$$\phi^0 : \mathbb{R}^d \times \mathbb{N} \to \mathbb{R}^m,$$
$$\phi^\ell : \mathbb{R}^m \times \mathbb{N} \to \mathbb{R}^m, \text{ for } \ell \in \{1, \ldots, L\},$$
$$\psi : \mathbb{R}^m \times \mathbb{N} \to \mathbb{R},$$

where for sequence $Y$ of length $N + N'$,

$$\phi^\ell(Y) = (\phi^\ell(y_1, 1), \ldots, \phi^\ell(y_{N+N'}, N + N')).$$

To evaluate $f(X)$ on some input sequence $X = (x_1, \ldots, x_N) \in \mathbb{R}^{N \times d}$, we define intermediate embeddings $Y^0, \ldots, Y^L \in \mathbb{R}^{(N+N') \times m}$ as follows:

- The initial embedding is computed by applying the first MLP $\phi^0$ to all elements of the input $X$, along with $N'$ "blank" inputs:

$$Y^0 = \phi^0(X) = (\phi^0(x_1, 1), \ldots, \phi^0(x_N, N), \phi^0(0, N + 1), \ldots, \phi^0(0, N + N')).$$

- The intermediate embeddings $Y^\ell$ for $\ell \in \{1, \ldots, L\}$ are computed by applying a unit of *multi-headed attention* (parameterized by $Q_{\ell,h}, K_{\ell,h}, V_{\ell,h}$ for all $h \in [H]$) to the previous embedding $Y^{\ell-1}$, followed by an element-wise application of the MLP $\phi^\ell$. That is,

$$Y^\ell = \phi^\ell \left( \sum_{h=1}^{H} \mathrm{softmax}\left( \phi(Y^{\ell-1}) Q_h K_h^\mathsf{T} \phi(Y^{\ell-1})^\mathsf{T} \right) \phi(Y^{\ell-1}) V_h \right).$$

---

[5]If $N' = 0$, we denote the family as $\mathsf{Transformer}_{m,H,L}^{N}$.

- The output $f(X) \in \mathbb{R}^N$ is computed to be the first $N$ outputs of $\psi(Y^L)$. That is,

$$f(X) = (\psi(Y_1^L, 1), \dots, \psi(Y_N^L, N)).$$

While our theoretical results pertain to the bidirectional regime (where no masking is applied to the self-attention layer), all results in Appendix C and all lower bounds in Appendix B apply to autoregressive transformers as well. Our empirical results utilize causal masking.

Furthermore, an $L$-layer bidirectional transformer that operates on sequences of length $N$ with $N'$ pause tokens can be simulated by and $L$-layer autoregressive transformer with $O(L \cdot (N + N'))$ pause tokens. In the regime where $N' = \text{poly}(N)$ and $L = O(\log N)$, any positive results for bidirectional models likewise apply to autoregressive models.

## A.2 Graph reasoning tasks

We exclusively consider graph reasoning tasks that apply to undirected and unweighted graphs $G = (V, E)$ of bounded size, i.e. $|V| + |E| = O(N)$ for some size parameter $N$. We define the tasks used for experimentation as follows:

- **Node count**: Given graph $G$, compute $|V|$.
- **Edge count**: Given graph $G$, compute $|E|$.
- **Edge existence**: Given graph $G$ and vertices $u, v \in V$, return $1\{(u, v) \in E\}$.
- **Node degree**: Given graph $G$ and vertex $u \in V$, return $\deg(u) = |(u, v) \in E|$.
- **Connectivity**: Given graph $G$ and vertices $u, v \in V$, return 1 if there exists a path of edges between $u$ and $v$ and 0 otherwise.
- **Cycle check**: Given graph $G$, return 1 if there exists a cycle of any length $\geq 3$ and 0 otherwise.
- **Shortest path**: Given graph $G$ and vertices $u, v \in V$, return the smallest number of edges that forms a path between $u$ and $v$ if one exists and $-1$ otherwise.
- **Triangle counting**: Given graph $G$, return the number of triangles in the graph: $|\{(u, v, w) : (u, v), (v, w), (u, w) \in E\}|$.

These tasks are foundational to computer science and are solved by famous polynomial-time algorithms, such as Dijkstra's algorithm for shortest path and Kruskal's for spanning trees. Several of these tasks reap significant algorithmic benefits from parallel computing models, including graph connectivity, which can be evaluated in a logarithmic number of rounds by employing an "iterative merging" strategy [4].

The amenability of some of these tasks to parallel computing is the core principle underlying our theoretical results on transformer model capacity and the resulting contrasts with GNNs. For instance, the iterative merging algorithm for connectivity can be simulated by a logarithmic-depth transformer, but message-passing GNNs cannot do without a substantial width scaling (see Appendix D). Furthermore, this parallel computing algorithm is widely believed to be optimal, and a crystallization of this as Conjecture 13 is the bedrock of our transformer optimality conjectures.

The multi-layer results of Appendix B are representation-agnostic; the bounds apply to any fixed encoding of vertices and edges as a transformer input sequence $X \in \mathbb{R}^{N \times d}$. The input representation for Appendix C must satisfy the more specified *node/edge encoding scheme*, which represents each vertex and edge as a single embedding, followed by a query embedding (with optional blank embeddings as needed). This encoding scheme is reflected by Figure 1 and closely resembles the graph instance encoding used for the experimental results[6].

Throughout the paper, we partition these and other graph reasoning tasks into three representational categories: retrieval, parallelizable, and search tasks. The retrieval category contains *only* four of the tasks used for experiments:

---

[6]The empirical encoding scheme uses two tokens—not one—to encode each edge and has fixed size $N$. For all graphs whose encodings do not require all $N$ tokens, the remaining tokens are blank placeholders that precede the query tokens.

- **Retrieval tasks** include node count, edge count, edge existence, and node degree.

On the other hand, the other categories reflect two equivalence classes introduce by [60]:

- **Parallelizable tasks** are defined to be L-complete and equivalent to connectivity in $O(1)$ rounds of MPC. These tasks include connectivity, cycle check, planarity testing, minimum cut, bipartiteness testing, minimum spanning forest, connected components, one-cycle versus two-cycles (see Conjecture 13), and # connected components.
- **Search tasks** are defined to be NL-complete and equivalent to shortest path in $O(1)$ rounds of MPC. These tasks include shortest path, strong connectivity (for directed graphs), all-pairs shortest path, median, diameter, radius, directed cycle detection, and $st$-reachability.

Triangle counting does not appear in any of these classes and is separately analyzed in Section 3.4.

## B    Multi-layer transformers and parallelizable/search graph reasoning tasks

In this appendix, we formalize and prove all results in Sections 3.1 and 3.2 that pertain to the parallelizable and search categories of graph reasoning tasks.

The primary technical tool used to establish these results is an improved relationship between the MPC model of distributed communication of [39] and our theoretical model of a transformers (Appendix A.1). This result is presented informally as Theorem 1, and formally as Theorem 8. Because graph reasoning tasks are well studied in the MPC model of computation [e.g. 4, 60], this theorem enables the transfer of those positive results to transformer architectures. Similarly, negative results that pertain to the MPC model, including those conditioned on the well-known *one-cycle versus two-cycle conjecture* (Conjecture 13), imply negative results for transformers.

**Theorem 8** (Formal version of Theorem 1; transformers simulate MPC). *For constants $0 < \delta < \delta' < 1$ and $\gamma > 0$, an $R$-round deterministic $(\gamma, \delta)$-MPC protocol with $n$ inputs can be simulated by a transformer $f \in \mathsf{Transformer}_{m,H,L}^{N,N'}$ with depth $L = O(R)$, single heads $H = 1$, embedding dimension $m = O(n^{\delta'})$, context length $N = n$, and blank chain-of-thought tokens $N' = \max(0, O(n^{1+\gamma-\delta}) - n)$.*

We formally introduce the MPC distributed computing model in Appendix B.1, along with the one-cycle versus two-cycle conjecture and a collection of positive results from [60]. In Appendix B.2, we formally present the task-specific graph reasoning results introduced in Sections 3.1 and 3.2 and prove them as implications of Theorem 8. We finally contextualize and prove Theorem 8 in Appendix B.3 by modifying the proof strategy of a similar result [71] and by showing that MPC protocols can be simulated by a weaker model of distributed computation.

### B.1    MPC preliminaries

The massively parallel computation (MPC) model of [39] formalizes distributed computing frameworks such as MapReduce [19] as theoretical models that are amenable to rigorous analysis. MPC pertains to a regime where an input that consists of $n$ "words" is distributed across a very large number of machines $q$ (e.g. $q \approx n^{0.95}$), each of which contains a bounded local memory $s$ (e.g. $s \approx n^{0.1}$) where computation on individual machines is inexpensive but communication between machines is costly. We use the definition of MPC of [4], which quantifies the complexity of a protocol by the local memory $s = O(n^{\delta})$, the global memory $sq = O(n^{1+\gamma})$, and the number of communication rounds $R$.

**Definition 2.** For global and local memory constants $\gamma, \delta > 0$ and input size $n$, an $R$-round $(\gamma, \delta)$-MPC protocol specifies a distributed computing protocol over $q = \Theta(n^{1+\gamma-\delta})$ machines, each of which contains a local memory of size $s = O(n^{\delta})$.

- An input to the protocol, which is encoded as a length-$n$ sequence of $p = \Theta(\log n)$-bit words, is distributed across the local memories of the first $\lceil n/s \rceil$ machines.
- In each of the $R$ rounds, each machine computes an arbitrary function its local memory at the time, which specifies at most $s$ words to be transmitted to other machines.
- Messages are simultaneously transmitted (subject to the constraint that each machine sends and receives at most $s$ words of information), and each machine's new local memory is set equal to the messages received.

- After the final round, the output of the protocol is a concatenation of the local memories of the first $\lceil n/s \rceil$ machines.

We say that an MPC protocol computes some function $f : \mathbb{Z}_{2^p}^n \to \mathbb{Z}_{2^p}^n$ if for any $X \in \mathbb{Z}_{2^p}^n$ encoded as input to the protocol, the output of the protocol is $f(X)$.

### B.1.1 MPC and graph reasoning tasks

In this section, we introduce previously known results about the abilities of Massively Parallel Computation protocols to solve graph reasoning tasks. The novel results presented in the subsequent section are the primarily the immediate implications of these results and Theorem 8.

**MPC round equivalence for parallelizable and search tasks**  These results largely reflect the contributions of [60], which establish a hierarchy of graph reasoning tasks that is reflected in our distinction between parallelizable and search tasks. They show that all parallelizable tasks are equivalent to connectivity in $O(1)$ rounds of MPC and that all search tasks are likewise equivalent to shortest path. (See their Figure 1.) Concretely, they prove the following for all graphs $G = (V, E)$ with $|V| + |E| = O(n)$.

**Theorem 9** (Theorem 4 of [60]; round-equivalence of parallelizable tasks). *If there exists an R-round $(\gamma, \delta)$-MPC protocol for any parallelizable task—including graph connectivity, $st$-connectivity, cycle detection, formula evaluation, planarity testing, graph bipartiteness, connected components, minimum spanning forest, and minimum cut, one-cycle versus two-cycle testing—for some $\gamma > 0$ and $\delta \in (0, 1)$, then there exists an $R'$-round $(\gamma', \delta)$-MPC protocol for any other other parallelizable task where $R' = R + O(1)$ and $\gamma' = \max(\gamma, 3)$.*

**Theorem 10** (Theorem 5 of [60]; round-equivalence of search tasks). *If there exists an R-round $(\gamma, \delta)$-MPC protocol for any search task—including $st$-reachability (for directed graphs), strong connectivitity, directed cycle detection, shortest path, all-pairs shortest path, diameter, radius, and median—for some $\gamma > 0$ and $\delta \in (0, 1)$, then there exists an $R'$-round $(\gamma', \delta)$-MPC protocol for any other other search task where $R' = R + O(1)$ and $\gamma' = \max(\gamma, 2)$.*

The equivalance of Theorem 9 has more immediate implications for the round complexity of parallelizable tasks than Theorem 10 does for search tasks because the round complexity of graph connectivity is well-understood to be $O(\log n)$ and thought to be optimal. We first present a deterministic MPC positive result for graph connectivity.

**Theorem 11** (Theorem 6.2 of [16]; log-round connectivity MPC algorithm). *For any $\gamma > 0$ and $\delta \in (0, 1)$, there exists a deterministic $O(\log n)$-round $(\gamma, \delta)$-MPC protocol that solves graph connectivity on any graph $G = (V, E)$ of size $|V| + |E| = O(n)$.*

Hence, all parallelizable tasks can be solved with a logarithmic number of MPC rounds with arbitrarily small polynomial local memory.

**Corollary 12** (Log-round parallelizable MPC algorithms). *For any $\gamma > 0$ and $\delta \in (0, 1)$ and any parallelizable task, there exists a deterministic $O(\log n)$-round $(\gamma', \delta)$-MPC protocol that solves the task on any graph $G = (V, E)$ of size $|V| + |E| = O(n)$ for $\gamma' = \max(\gamma, 3)$.*

The optimality of these logarithmic-round protocols is suggested by a conjecture about the hardness of distinguishing between two graphs with $n$ vertices and $n$ edges, one of whose edges are arranged in a single cycle of length $n$ and the other in two disjoint cycles of length $\frac{n}{2}$. This is the well-known "one-cycle versus two-cycle conjecture," which is widely employed as a condition for distributed computing hardness [see e.g. 8, 69, 27].

**Conjecture 13** (One-cycle versus two-cycle conjecture, see e.g. [27]). *For any $\gamma > 0$ and $\delta \in (0, 1)$, any $(\gamma, \delta)$-MPC protocol that distinguishes a single length-$n$ cycle from two disjoint length-$\frac{n}{2}$ cycles uses $R = \Omega(\log n)$ rounds.*

Under this conjecture, Theorem 9 immediately implies the optimality of Corollary 12.

**Corollary 14** (Optimality of log-round parallelizable MPC algorithms). *Conditional on Conjecture 13, for any $\gamma > 0$ and $\delta \in (0, 1)$, any $(\gamma, \delta)$-MPC protocol that solves a parallelizable graph task on all graphs $G = (V, E)$ of size $|V| + |E| = O(n)$ uses $R = \Omega(\log n)$ rounds.*

The round complexity of search tasks is less understood, and it is unknown if search tasks can be solved by $O(\log n)$-round $(\gamma, \delta)$-MPC protocols if $\delta \in (0, \frac{1}{2}]$.

**MPC protocols for problems with bounded circuit size**    More general MPC constructions are known for problems that solved by bounded-size boolean circuits, which include both parallelizable and search tasks. The well-known $\mathsf{NC}^i$ classes of Boolean circuits that take $N$ inputs and have $\mathrm{poly}(n)$ gates and depth $O(\log^i n)$ have been shown to be computable by bounded-round MapReduce-like computational protocols [25] and by MPC protocols in particular [60].

**Theorem 15** (Theorem 1 of [60]; log-round circuit MPC algorithms). *For any problem in $\mathsf{NC}^{i+1}$ and any $\gamma > 0$ and $\delta \in (\frac{1}{2}, 1)$, there exists a deterministic $O(\log^i n)$-round $(\gamma, \delta)$-MPC protocol that solves the problem.*

Since $\mathsf{L}$ and $\mathsf{NL}$ are known to belong to $\mathsf{NC}^2$, the following corollary is an immediate consequence.

**Corollary 16** (Log-round parallelizable and search MPC algorithms with high local memory). *For any parallelizable or search graph reasoning task and any $\gamma > 0$ and $\delta \in (\frac{1}{2}, 1)$, there exists a deterministic $O(\log n)$-round $(\gamma, \delta)$-MPC protocol that solves the task on all graphs $G = (V, E)$ of size $|V| + |E| = O(N)$.*

Note that these results pertain exclusively to the "large local memory regime," where each machine has memory at $s = \omega(\sqrt{n})$. Therefore, this does not guarantee the existence of a $O(\log n)$-round MPC solution for any search task or for any parallelizable task with $\gamma < 1$.

**MPC protocol for triangle counting**    Finally, the triangle counting task can be solved in the MPC framework by utilizing a special case of a parallel algorithm for clique counting. These pertain to graphs with bounded *arboricity* $\alpha$, a quantity that corresponds to the branching factor of a node that is bounded by the degree of the graph; these apply to arbitrary graphs by noting that $\alpha \leq |V|$.

**Theorem 17** (Theorem 1.5 of [11]; loglog-round triangle counting MPC algorithm). *For any $k \geq 3$, $\delta \in (0, 1)$, and $\gamma > 0$, there exists an $O(\log \log n)$-round $(\gamma, \delta)$-MPC protocol that computes the number of $k$-cliques in any graph $G = (V, E)$ with $|V| + |E| = O(n)$ and arboricity $\alpha = O(n^{\gamma/(k-2)})$.*

## B.2    Positive and negative graph reasoning results for multi-layer transformers

We state formalized versions of the statements of Sections 3.1, 3.2 and 3.4 and prove them by invoking Theorem 8 jointly with the relationships between MPC and graph reasoning of Appendix B.1.1.

### B.2.1    Parallelizable task results (Section 3.1)

For the duration of this section, the class of parallelizable tasks includes all of those that are deemed equivalent to graph connectivity in $O(1)$ rounds of MPC by [60], as stated in Theorem 9.

We first present a statement that formalizes the existence of logarithmic-depth transformer constructions for solving parallelizable tasks.

**Theorem 18** (Formal version of Theorem 2; log-depth transformers compute parallelizable tasks). *For any $\epsilon \in (0, 1)$ and any parallelizable task, there exists a transformer $f \in \mathsf{Transformer}_{m,H,L}^{N,N'}$ such that $f(X)$ computes the task where $X \in \mathbb{R}^N$ is some encoding of any graph $G = (V, E)$ with $|V| + |E| = O(N)$ and $f$ has depth $L = O(\log N)$ and heads $H = O(1)$ and embedding dimension $m$ and pause tokens $N'$ satisfying either*

- *LogDepthPause: $m = O(N^\epsilon)$ and $N' = O(N^{4-\epsilon'})$, where $\epsilon' \in (0, \epsilon)$; or*
- *LogDepthWide: $m = O(N^\epsilon)$ and $N' = 0$, if $\epsilon > \frac{1}{2}$.*

*Proof.* The first bullet is an immediate implication of Corollary 12 and Theorem 8 with $\gamma = 3$, $\delta' = \epsilon$, and $\delta = \epsilon'$. The second bullet follows from Corollary 16 and Theorem 8 with $\gamma = \delta = \frac{\epsilon}{2}$ and $\delta' = \epsilon$. $\square$

We then establish that sub-logarithmic-depth solutions to any parallelizable task are impossible without having linear embedding dimension $m$ or super-polynomial number of pause tokens $N'$, under the assumption that the one-cycle versus two-cycle conjecture holds.

**Theorem 19** (Formal version of Theorem 3; log-depth optimality for parallelizable tasks). *Conditional on Conjecture 13, for any $\epsilon \in (0, 1)$ and $\gamma > 0$ and any parallelizable task, if there exists a*

transformer $f \in \mathsf{Transformer}_{m,H,L}^{N,N'}$ that solves the task and has width $mH = O(N^\epsilon)$ and pause tokens $N + N' = O(N^{1+\gamma})$, then its depth satisfies $L = \Omega(\log N)$.

*Proof.* The proof is a consequence of Corollary 14 and a result of [71] that proves the to simulate transformers with MPC protocols, an inversion of Theorem 8. We restate that result as follows.

**Theorem 20** (Theorem 3.4 of [71]; MPC simulates transformers). *Fix any transformer $f \in$ $\mathsf{Transformer}_{m,H,L}^{N,N'}$ with width $mH = O(N^\delta)$ for some $\delta \in (0,1)$ and total sequence length $N + N' = O(N^{1+\gamma})$ for some $\gamma \geq 0$. Then, for any $\delta' \in (\delta, 1)$ and $\gamma' = 1 + 2\gamma + \delta'$, there exists an $O(\frac{L(1+\gamma)}{\delta'-\delta})$-round $(\gamma', \delta')$-MPC protocol that simulates $f$.*

Consider a transformer $f \in \mathsf{Transformer}_{m,L,H}^{N,N'}$ that solves the task with width $mH = O(N^\epsilon)$ and total sequence length $N + N' = O(N^{1+\gamma}$ for some constant $\epsilon \in (0,1)$ and $\gamma > 1$. Then, there exists an $O_{\epsilon,\gamma}(L)$-round $(1 + 2\gamma + \sqrt{\epsilon}, \sqrt{\epsilon})$-MPC protocol that solves the parallelizable task. If Conjecture 13 is true, then $L = \Omega(\log N)$. $\qquad\square$

### B.2.2 Search task results (Section 3.2)

We present the main result of Section 3.2 and show an equivalence between the minimum depth transformer needed to solve search tasks in a width-efficient manner. As before, the class of searchable tasks includes tasks that are equivalent to shortest path in $O(1)$ MPC rounds (Theorem 10).

**Theorem 21** (Formal version of Theorem 4; LogDepthWide computes search tasks). *For any $\epsilon \in (\frac{1}{2}, 1)$ and any search task, there exists a transformer $f \in \mathsf{Transformer}_{m,H,L}^{N,N'}$ such that $f(X)$ computes the task where $X \in \mathbb{R}^N$ is some encoding of any graph $G = (V, E)$ with $|V| + |E| = O(N)$, and $f$ has depth $L = O(\log N)$, heads $H = O(1)$, embedding dimension $m = O(N^\epsilon$, and no pause tokens ($N' = 0$).*

*Proof.* As before, the proof is an immediate consequence of Corollary 16 and Theorem 8. $\qquad\square$

While both search and parallelizable tasks are computable logarithmic depth of considerable width, the minimum depth transformer of width $O(N^\epsilon)$ for small $\epsilon$ that computes search tasks is unknown. Despite this deficit, we can still establish the representation similarity of all search tasks by showing that their minimum depths vary by at most a constant additive factor.

**Theorem 22** (Depth-equivalence of search tasks). *Suppose for some $\epsilon \in (0,1)$ and $\gamma > 0$ there exists a transformer $f \in \mathsf{Transformer}_{m,H,L}^{N,N'}$ with embedding dimension $m = O(N^\epsilon)$ and total sequence length $N + N' = O(N^{1+\gamma})$ that computes the some search task on all graphs $G = (V, E)$ of size $|V| + |E| = O(N)$. Then, for any other search task, there exists some transformer $f' \in$ $\mathsf{Transformer}_{\bar{m},H,\bar{L}}^{N,\bar{N}'}$ with embedding dimension $\bar{m} = O(m)$, depth $\bar{L} = L + O(1)$, and pause tokens $\bar{N}' = N' + O(N^3)$ that computes that search task.*

*Proof.* This statement is an immediate consequence of Theorem 10 and Theorem 8. $\qquad\square$

### B.2.3 Clique counting task results (Section 3.4)

We prove the existing of doubly-logarithmic-depth transformer that solves the triangle counting task giving a more general result that counts the number of $k$-cliques in any graph of bounded arboricity.

**Theorem 23** (Generalization of Theorem 7; loglog-depth computes clique counting). *For any fixed $\epsilon \in (0,1)$ and $k \geq 3$, there exists a transformer $f \in \mathsf{Transformer}_{m,H,L}^{N,N'}$ with embedding dimension $m = O(n^\epsilon)$, heads $H = O(1)$, depth $L = O(\log \log n)$, and chain-of-thought tokens*

$$N' = \begin{cases} O(\alpha^{k-2} N^{1-\epsilon}) & \text{if } \alpha = \Omega(N^{\epsilon/(k-2)}), \\ 0 & \text{otherwise.} \end{cases}$$

*that counts the number of $k$-cliques in all graphs $G = (V, E)$ of arboricity $\alpha$ and size $|V| + |E| = O(N)$.*

Theorem 7 follows immediately from taking $k = 3$.

*Proof.* This is the immediate implication of Theorem 8 and Theorem 23. □

## B.3 Proof of Theorem 8

Our principal theoretical result establishes that any MPC protocol with sublinear local memory can be simulated by a transformer with sublinear embedding dimension.

**Theorem 8** (Formal version of Theorem 1; transformers simulate MPC). *For constants $0 < \delta < \delta' < 1$ and $\gamma > 0$, an $R$-round deterministic $(\gamma, \delta)$-MPC protocol with $n$ inputs can be simulated by a transformer $f \in \mathsf{Transformer}_{m,H,L}^{N,N'}$ with depth $L = O(R)$, single heads $H = 1$, embedding dimension $m = O(n^{\delta'})$, context length $N = n$, and blank chain-of-thought tokens $N' = \max(0, O(n^{1+\gamma-\delta}) - n)$.*

This offers an improvement on Theorem 3.1 of [71], which only guarantees that MPC protocols with local memory $s = O(n^{1/4-\epsilon})$ (or with $\delta < \frac{1}{4}$) can be simulated by transformers with sublinear embedding dimension. This distinction is significant because several positive results for the MPC protocol (e.g. the ability to solve all problems in L and NL in a logarithmic number of MPC rounds) require $s = \Omega(N^{1/2})$ local memory, and hence could not be shown to be simulated by transformers of sublinear embedding dimension previously[7].

Including an allowance of $N'$ blank pause tokens permits the simulation of MPC protocol with $\gamma \geq \delta$, i.e. where number of machines $q$ grows super-linearly with $n$. If $\gamma < \delta$, then the protocol can be simulated without pause tokens (i.e. $N' = 0$) for sufficiently large input sizes $n$.

To prove Theorem 8, we define a restriction of the MPC computational model that disallows each machine from communicating with more than $k$ other machines in each round of the protocol.

**Definition 3.** For constants $\gamma, \delta, \rho > 0$, an $R$-round $(\gamma, \delta, \rho)$-*MPC protocol* on input sequences of length $n$ is a $(\gamma, \delta)$-MPC protocol (Definition 2) that obeys an additional constraint on the number of outgoing and incoming messages. Namely, for some capacity $k = O(n^\rho)$, in each round, every machine can send its local memory to and receive information from at most distinct $k$ machines.

Theorem 8 is an immediate consequence of Proposition 24—which establishes that any $(\gamma, \delta)$-MPC protocol can be simulated by a $(\gamma, \delta, \rho)$-MPC protocol—and Corollary 25—which shows that any $(\gamma, \delta, \rho)$-MPC protocol can be simulated by a transformer.

**Proposition 24** ($(\gamma, \delta, \rho)$-MPC simulates $(\gamma, \delta)$-MPC). *For constants $\gamma, \delta > 0$ and $\rho \in (0, \frac{\delta}{2})$, if $f$ can be computed by an $R$-round $(\gamma, \delta)$-MPC protocol, then there exists a $O(\frac{R(1+\gamma)^2}{\rho^2})$-round $(\gamma, \delta, \rho)$-MPC protocol that computes $f$ as well.*

**Corollary 25** (Transformers simulate $(\gamma, \delta, \rho)$-MPC). *For constants $\delta \in (0, 1)$ and $\gamma, \rho > 0$, an $R$-round deterministic $(\gamma, \delta, \rho)$-MPC protocol with $n$ inputs can be simulated by a transformer $f \in \mathsf{Transformer}_{m,H,L}^{N,N'}$ with depth $L = R+1$, heads $H = 1$, embedding dimension $m = O(n^{\delta+4\rho} \log n)$, context length $N = n$, and blank chain-of-thought tokens $N' = \max(0, O(n^{1+\gamma-\delta}) - n)$.*

*Proof of Theorem 8.* Let $\rho := \min(\frac{\delta}{2}, \frac{\delta'-\delta}{4}) - \epsilon$ for some small constant $\epsilon$ (e.g. $\epsilon := \min(\frac{\delta}{4}, \frac{\delta'-\delta}{8})$). By Proposition 24, we can simulate the target $R$-round $(\gamma, \delta)$-MPC protocol with an $R'$-round $(\gamma, \delta, \rho)$-MPC protocol for

$$R' = O\left(\frac{R(1+\gamma^2)}{\min((\delta'-\delta)^2, \delta^2)}\right).$$

We apply Corollary 25 to conclude that this latter protocol can be simulated by a transformer of depth $L = R' + 1$ and embedding dimension

$$m = O(n^{\delta+4\rho} \log n) = O(n^{\delta'}). \qquad \square$$

---

[7]For this theoretical model of the transformer to be non-vacuous, the transformer must have embedding dimension $m = o(Nd)$. Without this, any function over $X \in \mathbb{R}^{N \times d}$ could be solved by a single-layer transformer that concatenates all inputs into a single vector in an attention unit and passes that as input to an MLP that solves the problem.

We prove Proposition 24 in Appendix B.3.1 by simulating a single round of a standard MPC with multiple rounds of restricted MPC, where messages are sent to intermediate "neighborhoods"—which contain collections of machines with similar destinations—of increasing fine granularity. We prove Corollary 25 in Appendix B.3.2 by a minor adaptation of the proof of Theorem 3.1 of [71].

### B.3.1    Proof of Proposition 24

Fix an $R$-round $(\gamma, \delta)$-MPC protocol $\pi$ with input size $n$, $q = \Theta(n^{1+\gamma-\delta})$ machines, and $s = O(n^\delta)$ local memory. To prove Proposition 24, it suffices to show that a single round of MPC communication can be simulated an $O(\frac{(1+\gamma)^2}{\rho^2})$-round $(\gamma, \delta, \rho)$-MPC protocol.

To formalize the communication procedure to simulate, we let $\mathtt{Outbox} = (\mathtt{Outbox}_1, \ldots, \mathtt{Outbox}_q)$ denote an encoding of the outgoing "message packets" from each source machine that obeys $\pi$'s local memory constraints. Concretely,

$$\mathtt{Outbox}_i = \{(\mathtt{Msg}, \mathtt{Src}, \mathtt{Dst}) : \mathtt{Src} = i, \mathtt{Dst} \in [q]\} \text{ s.t.} \sum_{\mathtt{Msg} \in \mathtt{Outbox}_i} |\mathtt{Msg}| \leq s.$$

Let $\mathtt{Inbox} = (\mathtt{Inbox}_1, \ldots, \mathtt{Inbox}_q)$ denote the same collection of message packets, organized by their destination machines.

$$\mathtt{Inbox}_i = \{(\mathtt{Msg}, \mathtt{Src}, \mathtt{Dst}) \in \mathtt{Outbox}_{\mathtt{Src}} : \mathtt{Dst} = i\} \text{ s.t.} \sum_{\mathtt{Msg} \in \mathtt{Inbox}_i} |\mathtt{Msg}| \leq s.$$

It suffices to prove Lemma 26.

**Lemma 26** (($\gamma, \delta, \rho$)-MPC simulates one communication round of ($\gamma, \delta$)-MPC). *If $\rho < \frac{\delta}{2}$, there exists an $O(\frac{(1+\gamma)^2}{\rho^2})$-round $(\gamma, \delta, \rho)$-MPC protocol that takes as input any* $\mathtt{Outbox}$ *and returns the corresponding* $\mathtt{Inbox}$.

*Proof.* We define $r = O(\frac{1+\gamma}{\rho})$ intermediate steps and prove that each of those can be simulated. The intuition is that each an intermediate step routes each packet $(\mathtt{Msg}, \mathtt{Src}, \mathtt{Dst})$ to a machine that belongs to the same "neighborhood" as $\mathtt{Dst}$. Each step maps each packet to a neighborhood of smaller radius than the step before, until all packets have been transmitted to their proper destination location.

We define a tree of neighborhoods as follows. Fix some branching factor $\ell = \Theta(n^{\rho/2})$ and number of intermediate steps $r = \left\lceil \frac{\log q}{\log \ell} \right\rceil = O(\frac{1+\gamma}{\rho})$. For any step $t = 0, \ldots, r$ and neighborhood index $j = 1, \ldots, b_t := \left\lceil \frac{q}{\ell^{r-t}} \right\rceil$, we define

$$\mathtt{Nbhd}_j^t = \left\{(j-1)\ell^{r-t} + 1, (j-1)\ell^{r-t} + 2, \ldots, j \cdot \ell^{r-t}\right\} \cap [q]$$

and observe that it satisfies the size constraint $|\mathtt{Nbhd}_j^t| \leq \ell^{r-t}$. We let $\mathtt{Nbhd}^t(\mathtt{Dst})$ denote the unique neighborhood $\mathtt{Nbhd}_j^t$ satisfying $\mathtt{Dst} \in \mathtt{Nbhd}_j^t$. We say that $\mathtt{Nbhd}_j^t < \mathtt{Nbhd}_{j'}^t$ if $j < j'$ (and equivalently, for all $i \in \mathtt{Nbhd}_j^t$ and $i' \in \mathtt{Nbhd}_{j'}^t$, $i < i'$). Let

$$\mathtt{Children}(\mathtt{Nbhd}_j^t) = \left\{\mathtt{Nbhd}_{(j-1)\ell+1}^{t+1}, \ldots, \mathtt{Nbhd}_{j\ell}^{t+1}\right\},$$

$$\mathtt{Descendants}^\tau(\mathtt{Nbhd}_j^t) = \left\{\mathtt{Nbhd}^\tau(i) : i \in \mathtt{Nbhd}_j^t\right\},$$

$$\mathtt{Parent}(\mathtt{Nbhd}_j^t) = \mathtt{Nbhd}_{\lceil j/\ell \rceil}^{t-1},$$

$$\mathtt{Ancestor}^t(\mathtt{Nbhd}_j^\tau) = \mathtt{Nbhd}^t(i) \text{ for } i \in \mathtt{Nbhd}_j^\tau.$$

for some $\tau \geq t$.

Note that the following are true of neighborhoods.

- The initial "step 0" neighborhood contains all machines (i.e. $\mathtt{Nbhd}_1^0 = [q]$), and the final "step $r$" neighborhoods contain a single machine (i.e. $\mathtt{Nbhd}_j^r = \{j\}$ for all $j \in [q]$).

- For any $t < r$ and $j \in [b_t]$, $\mathtt{Nbhd}_j^t$ is the disjoint union of all sets in $\mathtt{Children}(\mathtt{Nbhd}_j^t)$ and $|\mathtt{Children}(\mathtt{Nbhd}_j^t)| \leq \ell$.

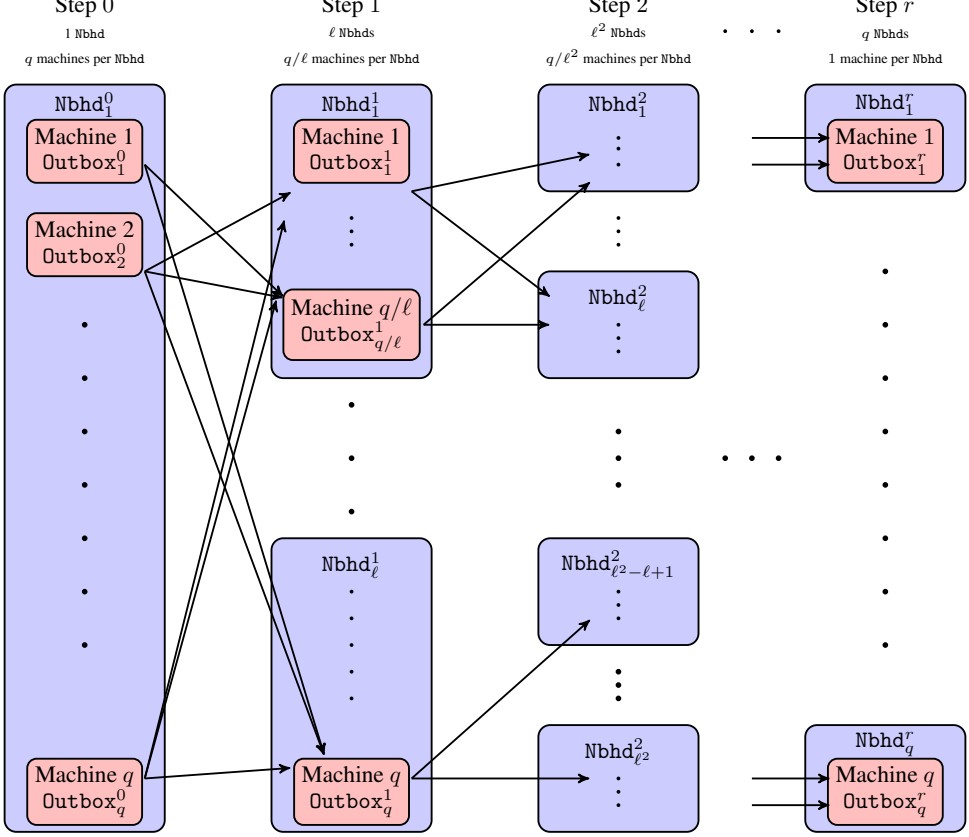

Figure 4: The neighborhood routing structure.

- $\left\{ \texttt{Nbhd}_1^t, \dots, \texttt{Nbhd}_{b_t}^t \right\}$ comprise a disjoint union of $[q]$.

- For any $\texttt{Dst} \in [q]$, there exist unique sets $\texttt{Nbhd}^0(\texttt{Dst}), \dots, \texttt{Nbhd}^r(\texttt{Dst})$ that satisfy

$$\texttt{Nbhd}^r(\texttt{Dst}) \subset \cdots \subset \texttt{Nbhd}^0(\texttt{Dst})$$

and $\texttt{Nbhd}^{t-1}(\texttt{Dst}) = \texttt{Parent}(\texttt{Nbhd}^t(\texttt{Dst}))$.

We define a collection of MPC machine states as "intermediate outboxes" $\texttt{Outbox}^t = (\texttt{Outbox}_1^t, \dots, \texttt{Outbox}_q^t)$ for each step $t$ to represent an assignment of message packets to machines in the same $t$th-level neighborhood as the packet destination. That is, we require that each $\texttt{Outbox}_i^t$ satisfy

$$\texttt{Outbox}_i^t = \left\{ (\texttt{Msg}, \texttt{Src}, \texttt{Dst}) : i \in \texttt{Nbhd}^t(\texttt{Dst}) \right\} \tag{1}$$

We say that $\texttt{Outbox}^t$ are *valid intermediate outboxes* for $\texttt{Outbox}$ if:

1. $\texttt{Outbox}^t$ satisfies Equation (1);

2. there is a one-to-one mapping between packets in $\texttt{Outbox}$ and $\texttt{Outbox}^t$; and

3. local memory constraints are maintained up to a factor of two[8]

$$\sum_{\texttt{Msg} \in \texttt{Outbox}_i^t} |\texttt{Msg}| \le 2s.$$

---

[8]For the sake of simplicity, we assume that the encoding $\texttt{Msg}$ also contains all relevant metadata about the packet $\texttt{Src}$ and $\texttt{Dst}$. That is, we only track the message size $|\texttt{Msg}|$.

Note that $\mathtt{Outbox}$ satisfies the conditions for $\mathtt{Outbox}^0$, and the only sequence of embeddings that satisfies the conditions for $\mathtt{Outbox}^r$ is $\mathtt{Inbox}$. An inductive argument that applies Lemma 27 $r$ times completes the proof. $\qquad\square$

**Lemma 27** ($(\gamma, \delta, \rho)$-MPC simulates one intermediate step). *For any $t \in \{0, \ldots, r-1\}$ and $\rho < \frac{\delta}{2}$, there exists an $O(\frac{1+\gamma}{\rho})$-round $(\gamma, \delta, \rho)$-MPC protocol that takes as input any satisfactory $\mathtt{Outbox}^t$ and returns some satisfactory $\mathtt{Outbox}^{t+1}$.*

*Proof.* The protocol operates in two phases:

1. The *pre-computation phase*, where relevant message size metadata is computed using $O(r - t)$ rounds of communication.

2. The *message-passing phase*, where all messages are propagated in $r - t$ rounds of communication in order to convert $\mathtt{Outbox}^t$ to $\mathtt{Outbox}^{t+1}$.

Since $r = O(\frac{1+\gamma}{\rho})$, the bound on total number of rounds is satisfied.

The algorithm maintains the invariant that all communication occurs within $t$-level neighborhoods $\mathtt{Nbhd}_j^t$ without any mixing between neighborhoods; this is possible because of the assumed validity of $\mathtt{Outbox}^t$, which implies that all messages whose packets appear in $\mathtt{Nbhd}_j^t$ of $\mathtt{Outbox}^t$ have ultimate destinations in $\mathtt{Nbhd}_j^t$. Concretely, if $(\mathtt{Msg}, \mathtt{Src}, \mathtt{Dst}) \in \mathtt{Outbox}_i^t$ and $i \in \mathtt{Nbhd}_j^t$, then $\mathtt{Dst} \in \mathtt{Nbhd}_j^t$.

We first explain the routing strategy for the message-passing phase by describing an itinerary of machines that each message packet will be routed to and proving that this itinerary meets the conditions needed to be executed by an $(r - t)$-round $(\gamma, \delta, \rho)$-MPC protocol. We then describe how the metadata necessary for the itinerary computations can be obtained during the pre-computation phase.

**Message-passing phase.** We first introduce some *packet* notation to track all relevant metadata about any message in the $t$th intermediate step. Fix some particular input $\mathtt{Outbox}^t$.

- Let $\mathtt{P} = (\mathtt{Msg}, \mathtt{Src}, \mathtt{Dst}, \mathtt{Src}^t)$ denote a *packet* that contains a message $\mathtt{Msg}$ and metadata concerning its "global" source $\mathtt{Src}$ and destination $\mathtt{Dst}$ (i.e. $(\mathtt{Msg}, \mathtt{Src}, \mathtt{Dst}) \in \mathtt{Outbox}_{\mathtt{Src}}, \mathtt{Inbox}_{\mathtt{Dst}}$) and its "local" source $\mathtt{Src}^t$ from within $\mathtt{Outbox}^t$ (i.e. $(\mathtt{Msg}, \mathtt{Src}, \mathtt{Dst}) \in \mathtt{Outbox}_{\mathtt{Src}^t}^t$).

- We write $\mathtt{P} \in \mathtt{Outbox}_i^t$ if $\mathtt{Src}^t = i$ and $\mathtt{P} \in \mathtt{Nbhd}$ if $\mathtt{P} \in \mathtt{Outbox}_i^t$ for some $i \in \mathtt{Nbhd}$.

- Let $\mathtt{SrcNbhd}^{t,\tau}(\mathtt{P}) = \mathtt{Nbhd}^\tau(\mathtt{Src}^t)$ represent the neighborhood of size $\ell^{r-\tau}$ contains $\mathtt{P}$ (i.e. $\mathtt{P} \in \mathtt{SrcNbhd}^{t,\tau}(\mathtt{P})$) and $\mathtt{DstNbhd}^t(\mathtt{P}) = \mathtt{Nbhd}^{t+1}(\mathtt{Dst})$ denote the neighborhood of size $\ell^{r-t-1}$ that contains the ultimate destination $\mathtt{Dst}$. Because $\mathtt{P} \in \mathtt{Nbhd}_j^t$ if and only if $\mathtt{Dst} \in \mathtt{Nbhd}_j^t$, it follows that $\mathtt{DstNbhd}^t(\mathtt{P}) \subset \mathtt{SrcNbhd}^{t,t}(\mathtt{P})$.

- Let $|\mathtt{P}| = |\mathtt{Msg}|$ be the total number of bits needed to encode the packet.

- We establish a lexicographical ordering that depends first on $\mathtt{Src}^t$ and next on $\mathtt{Dst}$. That is, we say that $P' < P$ for some $P' = (\mathtt{Msg}', \mathtt{Src}', \mathtt{Dst}', \mathtt{Src}^{t\prime})$ if $\mathtt{Src}^{t\prime} < \mathtt{Src}^t$, or if $\mathtt{Src}^{t\prime} = \mathtt{Src}^t$ and $\mathtt{Dst}' < \mathtt{Dst}$.

We develop a *packet scoring function* $z^{t,\tau}(\mathtt{P})$, which in turn induces an *itinerary function* $b^{t,\tau}(\mathtt{P}) \in [q]$ that assigns an intermediate machine to hold packet $\mathtt{P}$ after $r - \tau$ communication steps. These functions are carefully chosen in order to ensure that the local memories of each individual machine are bounded by $O(s)$ and that the number of machine each sends messages to and receives messages from is bounded by $O(\ell^2)$. Critically, we ensure that $b^{t,\tau}(\mathtt{P}) \in \mathtt{SrcNbhd}^{t,\tau}(\mathtt{P})$; that is, we initially rearrange packets within solely within the smallest neighborhoods of $\mathtt{Outbox}^t$ and gradually propagate packets more widely. That way, each packet $\mathtt{P}$ obeys a trajectory of the following form over $r - t$ MPC

rounds:

$$b^{t,r}(\mathtt{P}) = \mathtt{Src}^t \in \mathtt{SrcNbhd}^{t,r}(\mathtt{P}) \implies b^{t,r-1}(\mathtt{P}) \in \mathtt{SrcNbhd}^{t,r-1}(\mathtt{P}) \implies \dots$$
$$\implies b^{t,t+1}(\mathtt{P}) \in \mathtt{SrcNbhd}^{t,t+1}(\mathtt{P}) \implies b^{t,t}(\mathtt{P}) \in \mathtt{DstNbhd}^t(\mathtt{P}).$$

To define these functions, we first define partial sums of message sizes in order to later determine an itinerary of machines that $\mathtt{P}$ will be routed to in between $\mathtt{Src}^t$ and some destination machine in $\mathtt{DstNbhd}^t(\mathtt{P})$. The first term, $\mathtt{EqualDstSum}^{t,\tau}(\mathtt{P})$, sums the size of all "lesser" packets that share a destination neighborhood $\mathtt{DstNbhd}^t(\mathtt{P})$ and a $\tau$th level input neighborhood $\mathtt{SrcNbhd}^{t,\tau}(\mathtt{P})$:

$$\mathtt{EqualDstSum}^{t,\tau}(\mathtt{P}) = \sum_{\substack{\mathtt{P}' \in \mathtt{SrcNbhd}^{t,\tau}(\mathtt{P}) \\ \mathtt{DstNbhd}^t(\mathtt{P}') = \mathtt{DstNbhd}^t(\mathtt{P}) \\ \mathtt{P}' < \mathtt{P}}} |\mathtt{P}'|.$$

The second term, $\mathtt{LessDstSum}^{t,\tau}(\mathtt{P})$, sums the size of all packets that share an input neighborhood but have a "smaller" destination neighborhood than $\mathtt{P}$:

$$\mathtt{LessDstSum}^{t,\tau}(\mathtt{P}) = \mathtt{LessDstSum}^t(\mathtt{SrcNbhd}^{t,\tau}(\mathtt{P}), \mathtt{DstNbhd}^t(\mathtt{P})),$$

where

$$\mathtt{LessDstSum}^t(\mathtt{SrcNbhd}, \mathtt{DstNbhd}) = \sum_{\substack{\mathtt{P}' \in \mathtt{SrcNbhd} \\ \mathtt{DstNbhd}^t(\mathtt{P}') < \mathtt{DstNbhd}}} |\mathtt{P}'|.$$

We now define the packet scoring and itinerary functions for any $\tau \in \{t, \dots, r\}$:

$$z^{t,\tau}(\mathtt{P}) = \begin{cases} 2s \cdot \min \mathtt{SrcNbhd}^{t,\tau}(\mathtt{P}) + \mathtt{LessDstSum}^{t,\tau}(\mathtt{P}) + \mathtt{EqualDstSum}^{t,\tau}(\mathtt{P}) & \tau \geq t+1, \\ 2s \cdot \min \mathtt{DstNbhd}^t(\mathtt{P}) + 2 \cdot \mathtt{EqualDstSum}^{t,\tau}(\mathtt{P}) & \tau = t. \end{cases}$$

$$b^{t,\tau}(\mathtt{P}) = \left\lfloor \frac{z^{t,\tau}(\mathtt{P})}{2s} \right\rfloor.$$

We prove a series of claims to establish that the packet scoring and itinerary functions are properly defined.

**Claim 28** (Itinerary range). *For any $\tau \in \{t, \dots, r\}$, the itinerary function satisfies*

$$b^{t,\tau}(\mathtt{P}) \in \begin{cases} \mathtt{SrcNbhd}^{t,\tau}(\mathtt{P}) & \tau \geq t+1 \\ \mathtt{DstNbhd}^t(\mathtt{P}) & \tau = t. \end{cases}$$

*As an immediate consequence, $b^{t,r}(\mathtt{P}) = \mathtt{Src}^t$ for $\mathtt{P} = (\mathtt{Msg}, \mathtt{Src}, \mathtt{Dst}, \mathtt{Src}^t)$.*

*Proof.* We first bound the scoring function $z^{t,\tau}$. Note that

$$\mathtt{LessDstSum}^{t,\tau}(\mathtt{P}) + \mathtt{EqualDstSum}^{t,\tau}(\mathtt{P}) \leq \sum_{\mathtt{P}' \in \mathtt{SrcNbhd}^{t,\tau}(\mathtt{P})} |\mathtt{P}'| - |\mathtt{P}|$$
$$\leq |\mathtt{SrcNbhd}^{t,\tau}(\mathtt{P})| \cdot 2s - |\mathtt{P}|.$$

Therefore, for $\tau > t$,

$$z^{t,\tau}(\mathtt{P}) \in \left[ 2s \cdot \min \mathtt{SrcNbhd}^{t,\tau}(\mathtt{P}), 2s \cdot (\min \mathtt{SrcNbhd}^{t,\tau}(\mathtt{P}) + |\mathtt{SrcNbhd}^{t,\tau}(\mathtt{P})|) - |\mathtt{P}| \right], \quad (2)$$

and

$$b^{t,\tau}(\mathtt{P}) \in [\min \mathtt{SrcNbhd}^{t,\tau}(\mathtt{P}), \min \mathtt{SrcNbhd}^{t,\tau}(\mathtt{P}) + |\mathtt{SrcNbhd}^{t,\tau}(\mathtt{P})|),$$

which proves the first case of the claim. The second case follows by observing that

$$\mathtt{EqualDstSum}^{t,t}(\mathtt{P}) \leq s \cdot |\mathtt{DstNbhd}^t(\mathtt{P})| - |\mathtt{P}|.$$

Were it not true, there would exist at least one machine $i \in \mathtt{DstNbhd}^t(\mathtt{P})$ that must receive more an $s$ quantity of messages at the end of the entire round of the protocol $\pi$ (i.e. $\sum_{\mathtt{Msg} \in \mathtt{Inbox}_i} |\mathtt{Msg}| > s$), which contradicts the MPC assumption. $\qquad \square$

**Claim 29** (Gaps between scores). *If* $P_1 \neq P_2$ *and* $z^{t,\tau}(P_1) \leq z^{t,\tau}(P_2)$, *then*

$$z_{t,\tau}(P_1) + |P_1| \leq z^{t,\tau}(P_2).$$

*Proof.* First, let $\tau > t$. Consider the case where $\mathtt{SrcNbhd}^{t,\tau}(P_1) \neq \mathtt{SrcNbhd}^{t,\tau}(P_2)$. By Claim 28 and our assumption that $z^{t,\tau}(P_1) \leq z^{t,\tau}(P_2)$, it must be the case that $\mathtt{SrcNbhd}^{t,\tau}(P_1) < \mathtt{SrcNbhd}^{t,\tau}(P_2)$. Hence, we have the following by applying Equation (2).

$$z^{t,\tau}(P_2) - z_{t,\tau}(P_1)$$
$$\geq 2s \cdot (\min \mathtt{SrcNbhd}^{t,\tau}(P_2) - (\min \mathtt{SrcNbhd}^{t,\tau}(P_1) + |\mathtt{SrcNbhd}^{t,\tau}(P_1)| - |P_1|))$$
$$\geq |P_1|.$$

Otherwise, if $\mathtt{SrcNbhd}^{t,\tau}(P_1) = \mathtt{SrcNbhd}^{t,\tau}(P_2)$, then

$$z^{t,\tau}(P_2) - z_{t,\tau}(P_1) = (\mathtt{LessDstSum}^{t,\tau}(P_2) + \mathtt{EqualDstSum}^{t,\tau}(P_2)) - (\mathtt{LessDstSum}^{t,\tau}(P_1) + \mathtt{EqualDstSum}^{t,\tau}(P_1))$$
$$= \sum_{P' \in S_2} |P'| - \sum_{P' \in S_1} |P'|,$$

for some packet subsets $S_1, S_2 \subset \mathtt{SrcNbhd}^{t,\tau}(P_1)$. By further inspecting the respective $\mathtt{LessDstSum}^{t,\tau}$ and $\mathtt{EqualDstSum}^{t,\tau}$ terms, we observe that $S_1 \subset S_2$ and $P_1 \in S_2 \setminus S_1$. The claim immediately follows.

The argument for the case $\tau = t$ is nearly identical. □

**Claim 30** (Local memory bound). *For any* $b \in \mathbb{N}$,

$$\sum_{P : b^{t,\tau}(P)=b} |P| \leq \begin{cases} 2s & \tau \in \{t, r\} \\ 3s & \tau \in \{t+1, \ldots, r\}. \end{cases}$$

*Proof.* The case $\tau = r$ is an immediate consequence of the inductive assumption that $\mathtt{Outbox}^t$ satisfies the desired intermediate properties.

For all other cases, let $\{P_1, \ldots, P_n\}$ denote all packets with $b^{t,\tau}(P_i) = b$ and let $z^{t,\tau}(P_1) \leq \cdots \leq z^{t,\tau}(P_n)$ without loss of generality. We use Claim 29, the assumption that all $|P_i| \leq s$, and the boundedness of $z^{t,\tau}(P_i)$ from Claim 28 to conclude the proof.

$$\sum_{i=1}^{n} |P_i| \leq \sum_{i=1}^{n-1} (z^{t,\tau}(P_{i+1}) - z^{t,\tau}(P_i)) + |P_n|$$
$$\leq z^{t,\tau}(P_n) - z^{t,\tau}(P_1) + s$$
$$\leq \begin{cases} 2s & \tau = t, \\ 3s & \tau > t. \end{cases}$$
□

**Claim 31** (Intra-class distance preservation). *If* $P_1$ *and* $P_2$ *satisfy* $\mathtt{SrcNbhd}^{t,\tau+1}(P_1) = \mathtt{SrcNbhd}^{t,\tau+1}(P_2)$ *and* $\mathtt{DstNbhd}^t(P_1) = \mathtt{DstNbhd}^t(P_2)$, *then*

$$z^{t,\tau}(P_1) - z^{t,\tau}(P_2) = z^{t,\tau+1}(P_1) - z^{t,\tau+1}(P_2).$$

*Proof.* Since $\mathtt{SrcNbhd}^{t,\tau+1}(P_1) = \mathtt{SrcNbhd}^{t,\tau+1}(P_2)$, it follows that $\mathtt{SrcNbhd}^{t,\tau}(P_1) = \mathtt{SrcNbhd}^{t,\tau}(P_2)$ and therefore,

$$z^{t,\tau}(P_1) - z^{t,\tau}(P_2) = \mathtt{EqualDstSum}^{t,\tau}(P_1) - \mathtt{EqualDstSum}^{t,\tau}(P_2)$$

$$= \sum_{\substack{P' \in \mathtt{SrcNbhd}^{t,\tau}(P_1) \\ \mathtt{DstNbhd}^t(P')=\mathtt{DstNbhd}^t(P_1) \\ P_1 \leq P' < P_2}} |P'| \tag{3}$$

$$z^{t,\tau+1}(P_1) - z^{t,\tau+1}(P_2) = \mathtt{EqualDstSum}^{t,\tau+1}(P_1) - \mathtt{EqualDstSum}^{t,\tau+1}(P_2)$$

$$= \sum_{\substack{P' \in \mathtt{SrcNbhd}^{t,\tau+1}(P_1) \\ \mathtt{DstNbhd}^t(P')=\mathtt{DstNbhd}^t(P_1) \\ P_1 \leq P' < P_2}} |P'|. \tag{4}$$

Because $P_1, P_2 \in \texttt{SrcNbhd}^{t,\tau+1}(P_1)$, the defined packet ordering implies that any $P' \in [P_1, P_2)$ must satisfy $P' \in \texttt{SrcNbhd}^{t,\tau+1}(P_1)$. Therefore, Equations (3) and (4) are equal and the claim holds. $\quad\square$

**Claim 32** (Distinct recipients bound). *For any b,*
$$\left| \left\{ b^{t,\tau}(P) : b^{t,\tau+1}(P) = b \right\} \right| \leq 3\ell.$$

*Proof.* Within each root neighborhood $\texttt{Nbhd}_j^\tau$, there exist at most $\ell$ destination neighborhoods in $\texttt{DstNbhd}^t(P)$ for $P \in \texttt{Nbhd}_j^\tau$.

Fix some such $\texttt{DstNbhd}$ and let $P_1, \ldots, P_n$ denote all packets with $b^{t,\tau+1}(P) = b$ and $\texttt{DstNbhd}^t(P_i) = \texttt{DstNbhd}$. Without loss of generality, assume that $z^{t,\tau}(P_1) \leq \cdots \leq z^{t,\tau}(P_n)$. Because all such packets belong to the same machine in step $r - \tau - 1$ (i.e. $b^{t,\tau+1}(P_i) = b$ for all $i \in [n]$), they belong share the same source neighborhood of size $\ell^{r-\tau-1}$ (i.e. $\texttt{SrcNbhd}^{t,\tau+1}(P_i) = \texttt{SrcNbhd}^{t,\tau+1}(P_1)$). By Claim 31 and the definition of $b^{t,\tau}$,

$$b^{t,\tau}(P_i) - b^{t,\tau}(P_1) \leq 1 + \frac{1}{2s}(z^{t,\tau}(P_i) - z^{t,\tau}(P_1))$$
$$= 1 + \frac{1}{2s}(z^{t,\tau+1}(P_i) - z^{t,\tau+1}(P_1))$$
$$\leq 2 + b^{t,\tau+1}(P_i) - b^{t,\tau+1}(P_1) = 2.$$

Therefore, there are at most three possible values of $b^{t,\tau}(P_i)$.

The claim follows by considering each of the $\ell$ destination neighborhoods separately:

$$\left| \left\{ b^{t,\tau}(P) : b^{t,\tau+1}(P) = b \right\} \right| = \sum_{\texttt{DstNbhd}} \left| \left\{ b^{t,\tau}(P) : b^{t,\tau+1}(P) = b, \texttt{DstNbhd}^t(P) = \texttt{DstNbhd} \right\} \right|$$
$$\leq 3\ell. \qquad\qquad \square$$

**Claim 33** (Distinct senders bound). *For any b,*
$$\left| \left\{ b^{t,\tau+1}(P) : b^{t,\tau}(P) = b \right\} \right| \leq 3\ell^2.$$

*Proof.* Within each $\texttt{Nbhd}_j^\tau$, there exist at most $\ell^2$ distinct pairs of destination neighborhoods $\texttt{DstNbhd}^t(P)$ and source neighborhoods $\texttt{Nbhd}^{t,\tau}(P)$ for $P \in \texttt{Nbhd}_j^\tau$.

As before, we fix some $\texttt{DstNbhd}$ and $\texttt{SrcNbhd}$ and let $P_1, \ldots, P_n$ all satisfy $b^{t,\tau}(P_i) = b$, $\texttt{DstNbhd}^t(P_i) = \texttt{DstNbhd}$, and $\texttt{SrcNbhd}^{t,\tau+1} = \texttt{SrcNbhd}$. Using the same argument, we show that

$$\left| \left\{ b^{t,\tau+1}(P_i) : i \in [n] \right\} \right| \leq 3.$$

We conclude by considering all such pairs. $\quad\square$

As a result of Claims 30, 32, and 33, we conclude that each packet $P = (\texttt{Msg}, \texttt{Src}, \texttt{Dst}, \texttt{Src}^t)$ can be equipped with some itinerary

$$\texttt{Src}^t = b^{t,r}(P), \ b^{t,r-1}(P), \ldots, b^{t,t+1}(P), \ b^{t,t}(P) \in \texttt{DstNbhd}^t(P)$$

that properly translates an instances of $\texttt{Outbox}^t$ to $\texttt{Outbox}^{t+1}$ and does so without ever requiring local memory more than $3s = O(N^\delta)$ on any intermediate step or any machine to send or receive messages from more than $3\ell^2 = O(N^\rho)$ other machines. This itinerary can be executed using an $(r - t)$-round $(\gamma, \delta, \rho)$-MPC protocol.

**Pre-computation phase.** It remains to show that each $b^{t,\tau}$ can be computed for each packet. To do so, we prove that there exists an $O(r - t)$-round $(\gamma, \delta, \rho)$-MPC protocol that ends with each machine $i$ knowing $\texttt{EqualDstSum}^{t,\tau}(P)$ and $\texttt{LessDstSum}^{t,\tau}(P)$ for each $P \in \texttt{Outbox}_i^t$ and $\tau \in \{t, \ldots, r\}$. We design an MPC protocol that uses the tree structure to propagate information about individual child neighborhoods to their parents and vice versa. We describe recursive relationships that elucidate how to compute the two salient quantities.

First, we introduce a useful intermediate term. Let

$$\texttt{EqualDstSum}^{t,\tau}(i, \texttt{DstNbhd}) = \sum_{\substack{i' \in \texttt{Nbhd}^\tau(i) \\ i' < i}} \sum_{\substack{P' \in \texttt{Outbox}_{i'}^t \\ \texttt{DstNbhd}^t(P') = \texttt{DstNbhd}}} |P'|,$$

denote the sum of all packet that are contained by "lesser" machines that share source and destination neighborhoods. Note that $\mathtt{EqualDstSum}^{t,\tau}(\mathtt{P})$ can be computed for any $\mathtt{P} \in \mathtt{Outbox}_i^t$ locally by machine $i$ given prior knowledge of $\mathtt{EqualDstSum}^{t,\tau}(i, \mathtt{DstNbhd}^t(\mathtt{P}))$. Thus, the pre-computation phase need only compute the latter term.

We also introduce a term that represents sum of the sizes of all packets that share source and destination neighborhoods:

$$\mathtt{NbhdSum}^t(\mathtt{SrcNbhd}, \mathtt{DstNbhd}) = \sum_{\substack{\mathtt{P}' \in \mathtt{SrcNbhd} \\ \mathtt{DstNbhd}^t(\mathtt{P}') = \mathtt{DstNbhd}}} |\mathtt{P}'|.$$

Now, we provide the recurrences for any $\tau < r$ (or for any $\mathtt{SrcNbhd}$ satisfying $|\mathtt{SrcNbhd}| > 1$):

$\mathtt{EqualDstSum}^{t,\tau}(i, \mathtt{DstNbhd})$
$\quad = \mathtt{EqualDstSum}^{t,\tau+1}(i, \mathtt{DstNbhd}) + \mathtt{EqualDstSum}^{t,\tau}(\min \mathtt{Nbhd}^{\tau+1}(i), \mathtt{DstNbhd})$

$$= \mathtt{EqualDstSum}^{t,\tau+1}(i, \mathtt{DstNbhd}) + \sum_{\substack{\mathtt{SrcNbhd} \in \mathtt{Children}(\mathtt{Nbhd}^\tau(i)) \\ \mathtt{SrcNbhd} < \mathtt{Nbhd}^{\tau+1}(i)}} \mathtt{NbhdSum}^t(\mathtt{SrcNbhd}, \mathtt{DstNbhd}),$$

$$\mathtt{NbhdSum}^t(\mathtt{SrcNbhd}, \mathtt{DstNbhd}) = \sum_{\mathtt{SrcNbhd}' \in \mathtt{Children}(\mathtt{SrcNbhd})} \mathtt{NbhdSum}^t(\mathtt{SrcNbhd}, \mathtt{DstNbhd}), \quad (5)$$

$$\mathtt{LessDstSum}^t(\mathtt{SrcNbhd}, \mathtt{DstNbhd}) = \sum_{\mathtt{SrcNbhd}' \in \mathtt{Children}(\mathtt{SrcNbhd})} \mathtt{LessDstSum}^t(\mathtt{SrcNbhd}, \mathtt{DstNbhd}).$$

$$(6)$$

When $\tau = r$, the terms $\mathtt{EqualDstSum}^{t,\tau}(i, \mathtt{DstNbhd})$, $\mathtt{NbhdSum}^t(\mathtt{Nbhd}^r(i), \mathtt{DstNbhd})$, and $\mathtt{LessDstSum}^t(\mathtt{Nbhd}^r(i), \mathtt{DstNbhd})$ can be computed locally within machine $i$.

We follow a tree-like communication pattern to compute all relevant sums. Each machine $i \in [q]$ computes

$$\left\{ \mathtt{EqualDstSum}^{t,\tau}(i, \mathtt{DstNbhd}) : \tau \geq t, \mathtt{DstNbhd} \in \mathtt{Children}(\mathtt{Nbhd}^t(i)) \right\}$$

and

$$\left\{ \mathtt{LessDstSum}^t(\mathtt{Nbhd}^\tau(i), \mathtt{DstNbhd}) : \tau \geq t, \mathtt{DstNbhd} \in \mathtt{Children}(\mathtt{Nbhd}^t(i)) \right\}$$

by completing $r - t$ *propagate-up* rounds, $r - t$ *aggregation* rounds, and $r - t$ *propagate-down* rounds.

- The propagate-up rounds compute the neighborhood-wide message-size summations $\mathtt{NbhdSum}^t(\mathtt{Nbhd}^\tau(i), \mathtt{DstNbhd})$ and $\mathtt{LessDstSum}^t(\mathtt{Nbhd}^\tau(i), \mathtt{DstNbhd})$ in each machine $i$ satisfying $i = \min \mathtt{Nbhd}^\tau(i)$ for each $\tau$ and $\mathtt{DstNbhd}$.

- The aggregation rounds accumulate $\mathtt{NbhdSum}$ terms of the same level into specific $\mathtt{EqualDstSum}$ terms.

- The propagate-down rounds iteratively compute and share the $\mathtt{EqualDstSum}$ and $\mathtt{LessDstSum}$ terms with all relevant machines.

*Propagate-up rounds:* Fix some neighborhood $\mathtt{Nbhd}_j^t$. After $r - \tau$ propagate-up rounds, the goal is to compute the terms $\mathtt{NbhdSum}^t(\mathtt{SrcNbhd}, \mathtt{DstNbhd})$ and $\mathtt{LessDstSum}^t(\mathtt{SrcNbhd}, \mathtt{DstNbhd})$ for each relevant destination neighborhood $\mathtt{DstNbhd} \in \mathtt{Children}(\mathtt{Nbhd}_j^t)$ and source neighborhood $\mathtt{SrcNbhd} \in \mathtt{Descendants}^\tau(\mathtt{Nbhd}_j^t)$ within a single machine in each source neighborhood, $\min \mathtt{SrcNbhd}$. We argue that this is possible inductively.

Before performing any computation (that is, after "round 0"), each machine $i$ individually computes $\mathtt{NbhdSum}^t(\mathtt{Nbhd}^r(i), \mathtt{DstNbhd})$ and $\mathtt{LessDstSum}^t(\mathtt{Nbhd}^r(i), \mathtt{DstNbhd})$ by aggregating the messages encoded in its own representation of $\mathtt{Outbox}_i^t$.

We assume that the procedure works as specified for $r - \tau$ rounds of communication. Fix some $\mathtt{SrcNbhd}^{\tau-1} \in \mathtt{Descendants}^{\tau-1}(\mathtt{Nbhd}_j^t)$. Then, for every $\mathtt{SrcNbhd}^{\tau} \in \mathtt{ChildrenSrcNbhd}^{\tau-1}$, the quantities

$$\{\mathtt{NbhdSum}^t(\mathtt{SrcNbhd}^{\tau}, \mathtt{DstNbhd}) : \mathtt{DstNbhd} \in \mathtt{Children}(\mathtt{Nbhd}_j^t)\}$$

$$\cup \{\mathtt{LessDstSum}^t(\mathtt{SrcNbhd}^{\tau}, \mathtt{DstNbhd}) : \mathtt{DstNbhd} \in \mathtt{Children}(\mathtt{Nbhd}_j^t)\}$$

have already been computed and stored in $\min \mathtt{SrcNbhd}^{\tau}$. By the recurrence relations of Equations (5) and (6), $\mathtt{NbhdSum}^t(\mathtt{SrcNbhd}^{\tau-1}, \mathtt{DstNbhd})$ and $\mathtt{LessDstSum}^t(\mathtt{SrcNbhd}^{\tau-1}, \mathtt{DstNbhd})$ are functions of those quantities. Thus, it suffices to have each machine $\min \mathtt{SrcNbhd}^{\tau}$ machine transmit its relevant terms to $\min \mathtt{SrcNbhd}^{\tau-1}$. A round of MPC communication that transmits such messages involves each machine sending most one message of size $\ell$ and receiving at most $\ell$ messages, each of size $O(\ell)$.

Inductively, we ensure that all neighborhood sums are computed after $r - t$ propagate-up rounds. Because each machine handles at most $\ell$ distinct messages having total size $O(\ell^2)$ per MPC round, this protocol does not violate the bounded message size and bounded distinct message constraints (so long as $\ell^2 \ll s$), which can be guaranteed for sufficiently large $n$, so long as $\ell = O(n^{\rho/2})$.

*Aggregation rounds:* After the completion of the aggregation rounds, each machine $i$ computes terms of the form

$$\{\mathtt{EqualDstSum}^{t,\tau}(\min \mathtt{Nbhd}^{\tau+1}(i), \mathtt{DstNbhd}) : \mathtt{DstNbhd} \in \mathtt{Children}(\mathtt{Nbhd}_j^t)\}$$

from relevant $\mathtt{NbhdSum}^t$ terms if $i = \min \mathtt{Nbhd}^{\tau+1}(i)$. By the recurrence, it is sufficient for machine $i$ to following $\ell^2$ distinct terms

$$\{\mathtt{NbhdSum}^t(\mathtt{SrcNbhd}, \mathtt{DstNbhd}) : \mathtt{SrcNbhd} \in \mathtt{Children}(\mathtt{Nbhd}^{\tau}(i)), \mathtt{DstNbhd} \in \mathtt{Children}(\mathtt{Nbhd}_j^t)\}.$$

Since all machines $i$ already knows such terms for $\mathtt{SrcNbhd} = \mathtt{Nbhd}^{\tau+1}(i)$, it can obtain the remaining $\mathtt{NbhdSum}^t$ terms by simultaneously sharing information with its "cousin machines:" $\min \mathtt{SrcNbhd}$, for all $\mathtt{SrcNbhd} \in \mathtt{Children}(\mathtt{Nbhd}^{\tau}(i))$. This can be handled by a single round of communication where each "$(\tau + 1)$th-level neighborhood representative" machine forwards its sums sums to up to $\ell$ other first representatives, for a total messaging cost of $O(\ell^2)$.

We use $r - t$ separate rounds to repeat this process for each $\tau \geq t$.

*Propagate-down rounds:* It remains to compute each $\mathtt{EqualDstSum}^{t,\tau}(i, \cdot)$ and $\mathtt{LessDstSum}^t(\mathtt{Nbhd}^{\tau}(i), \cdot)$ term at each machine $i$. The relevant $\mathtt{LessDstSum}^t$ terms have already been computed by each respective $\min \mathtt{Nbhd}^{\tau}(i)$ machine and can be propagated to machine $i$ by using $r - t$ rounds of tree-propagation through intermediate $\min \mathtt{Nbhd}^{\tau'}(i)$ machines.

The same is possible for $\mathtt{EqualDstSum}^{t,\tau}$ terms, although the individual terms need to be added in order to follow the recurrence. We propagate the terms in the same way as the $\mathtt{LessDstSum}^t$ terms, but we take special care to carry out the extra additions. This can be accomplished simultaneously to the other propagate-down rounds. This protocol involves each first-child node sharing at most $\ell$ distinct messages, each of size at most $O((r - t)\ell)$. As before, every node sends and receives at most $O((r - t)\ell^2) \ll s$ words.

After these $3(r - t)$ rounds have elapsed, each machine $i$ computes $b^{t,\tau}(\mathtt{P})$ for each $\mathtt{P} \in \mathtt{Outbox}_i^t$. Using this itinerary, the machines routes tuples of the form

$$(\mathtt{P}, b^{t,r}(\mathtt{P}), \ldots, b^{t,t}(\mathtt{P}))$$

to the respective machine $b^{t,\tau}(\mathtt{P})$ in round $r - \tau$. Due to the arguments presented at the start of the section, this procedure terminates with each $(\mathtt{Msg}, \mathtt{Src}, \mathtt{Dst})$ tuple being held by some machine $i$ such that the resulting $\mathtt{Outbox}_i^{t+1}$ is valid, and the procedure involves at most $O(r - t) = O(\frac{1+\gamma}{\rho})$ rounds of $(\gamma, \delta, \rho)$-MPC computation. $\qquad \square$

### B.3.2 Proof outline of Corollary 25

**Corollary 25** (Transformers simulate $(\gamma, \delta, \rho)$-MPC). *For constants $\delta \in (0, 1)$ and $\gamma, \rho > 0$, an $R$-round deterministic $(\gamma, \delta, \rho)$-MPC protocol with $n$ inputs can be simulated by a transformer $f \in \mathsf{Transformer}_{m,H,L}^{N,N'}$ with depth $L = R+1$, heads $H = 1$, embedding dimension $m = O(n^{\delta+4\rho} \log n)$, context length $N = n$, and blank chain-of-thought tokens $N' = \max(0, O(n^{1+\gamma-\delta}) - n)$.*

The proof of Corollary 25 involves an adaptation to the proof of Theorem 3.1 of [71]. To avoid restating the proof in its entirety, we provide a brief outline of the proof of Theorem 3.1 and explain which modification is necessary.

Theorem 3.1 is a consequence of Lemmas B.4, B.5, and B.6 of [71], which establish that there exist single-layer transformers that simulate the initialization, a round of computation and communication, and the output formatting of any fixed $(\gamma, \delta)$-MPC protocol. The input and output steps of $(\gamma, \delta)$, and $(\gamma, \delta, \rho)$-MPC protocols are identical, only Lemma B.5 needs to be examined.

To simulate a single round of an MPC protocol with a transformer, all local computations are simulated in the element-wise multi-layer perceptron (MLP) units, and all communication is handled in a single multi-headed self-attention layer (Lemma B.7) Since $(\gamma, \delta, \rho)$-MPC protocols add no restrictions related to local computation, the former can be simulated in exactly the same manner with identical MLPs, and it remains to analyze the construction of Lemma B.7. We restate Lemma B.7 and provide an replacement lemma that suffices to prove Corollary 25.

**Lemma 34** (Lemma B.7 of [71]; multi-headed attention simulates MPC communication). *For any $R$-round MPC protocol with local memory $s$ and $q$ machines and any round $r \in [R-1]$, there exists a single-layer transformer $f \in \mathsf{Transformer}^{q,0}_{m,H,1}$ with $H = O(\log \log q)$ and $m = O(s^4 \log q)$, which, given as input a length-$q$ encoding of each machine's outgoing messages in round $r$, returns an encoding of each machine's incoming messages in round $r + 1$.*

**Lemma 35** (Single-headed attention simulates bounded-message MPC communication). *For any $R$-round MPC protocol with local memory $s$, $q$ machines, and a $k$-machine communication limit and any round $r \in [R-1]$, there exists a single-layer single-headed transformer $f \in \mathsf{Transformer}^{q,0}_{m,H,1}$ with $H = 1$ and $m = O(k^4 s \log q)$, which, given as input a length-$q$ encoding of each machine's outgoing messages in round $r$, returns an encoding of each machine's incoming messages in round $r + 1$.*

Lemma 35 is an immediate consequence of Lemma 3.2 of [71], their main technical result, which already applies to the regime with limits on the number of machines in communication.

By replacing Lemma 34 with Lemma 35, applying the remainder of the proof of Theorem 3.1 of [71], and letting $k = O(n^\rho)$, the proof of Corollary 25 is complete.

## C  Single-layer transformers and graph reasoning tasks

This appendix presents the results of Section 3.3, which separates the collection of graph reasoning tasks into those *retrieval tasks* that can be efficiently solved by single-layer parameter-efficient transformers—including node count, edge count, edge existence, and node degree—and those *parallelizable* or *search tasks* that require deeper constructions—including connectivity, shortest path, cycle check, and triangle count. Taken together, these results establish that the single-layer transformers of the Depth1 regime are capable of solving simple aggregation-based tasks, but that their known limitations in capacity as communication protocols of [70] apply to non-trivial graph reasoning tasks.

We specific a particular *node/edge encoding* of an input graph $G = (V, E)$ and a graph reasoning task using a consistent encoding scheme that closely resembles the encoding used in our graph reasoning experiments and those of [42]. This encoding is distinguished by the fact that each node and vertex of the graph $G$ is represented by exactly one token, rather than the pair of tokens utilized in our experiments. This choice ensures that any trivial pre-processing of graph inputs (e.g. using a positional embedding to associate each edge token pair) need not count towards the single-layer transformer model.

**Definition 4.** The *node/edge encoding* of a graph $G = (V, E)$ with $V \subseteq [n]$ and $|V| + |E| \leq N - 1$ and a graph reasoning task $P$ is a sequence

$$X = X(G, P) = (x_1, \ldots, x_N) \in \mathbb{R}^{N \times d}$$

where $d = 5$ and each

$$x_i = (x_i^1, x_i^2, \mathtt{isVertex}_i, \mathtt{isEdge}_i, \mathtt{isTask}_i) \in \{0, \ldots, n\}^2 \times \{0, 1\}^3$$

satisfies the following conditions:

- For each $v \in V$, there exists exactly one $i \in [N-1]$ with $x_i = (v, 0, 1, 0, 0)$.

- For each $(u, v) \in E$, there exists exactly $i \in [N-1]$ with $x_i = (u, v, 0, 1, 0)$.

- The token $x_N$ encodes a particular instance of the task $P$, by encoding $\mathtt{isTask}_N = 1$ with an optional edge or node encoding. That is, for tasks without arguments (such as triangle count), $x_N = (0, 0, 0, 0, 1)$. For tasks with a single node argument $v \in [n]$ (such as node degree), $x_N = (v, 0, 1, 0, 1)$. For tasks with a pair of node arguments $u, v \in [n]$ (such as shortest path and connectivity), $x_N = (u, v, 0, 1, 1)$.

- All other tokens satisfy $x_i = (0, 0, 0, 0, 0)$.

We say that a single-layer transformer $f \in \mathsf{Transformer}_{m,H,1}^N$ *solves* task $P$ on graph $G$ if the output corresponding to the task description $f(X(G, P))_N$ encodes the output of the task. Since $f$ is a single-layer transformer, we can write this output as

$$f(X)_N = \psi \left( \sum_{h=1}^H \mathrm{softmax}\left( \phi(x_N)^\mathsf{T} Q_h K_h^\mathsf{T} \phi(X)^\mathsf{T} \right) \phi(X) V_h \right) \tag{7}$$

for element-wise multi-layer perceptrons

$$\phi : \mathbb{R}^d \to \mathbb{R}^m, \psi : \mathbb{R}^m \to \mathbb{R}$$

broadcasted across each input (i.e. $\phi(X) = (\phi(x_1), \dots, \phi(x_N)) \in \mathbb{R}^{N \times m}$) and weight matrices

$$Q_1, \dots, Q_H, K_1, \dots, K_H, V_1, \dots, V_H \in \mathbb{R}^{m \times m}.$$

Throughout, we assume that all parameters in the transformer model and intermediate numerical quantities can be written using $O(\log N)$-bit floating point numbers. This assumption can be satisfied for the positive results of Appendix C.1 and is necessary to obtain the negative results of Appendix C.2.

We permit the element-wise MLPs $\phi$ and $\psi$ to be arbitrary functions for the negative results, while restricting them to be MLPs that can be approximated using bounded-size multi-layer ReLU networks for the positive results. While we do not make these ReLU networks explicit, we restrict ourselves to simple operations that can be computed using linear transformations and the application of smooth univariate functions.

Finally, we acknowledge that the negative results in are largely superseded by those of [55], which establishes that L-complete languages (including graph connectivity and cycle check) cannot be efficiently solved by constant-depth transformers, let alone single-layer transformers. We include these bounds anyway to mark a contrast with our positive results; draw further connections to the communication complexity lens on transformers; and exhibit simple task instances $(G, P)$ that require greater depth, including constant diameter and constant degree graphs.

### C.1 Positive results for single-layer transformers

**Theorem 36** (Formal version of Theorem 5; Depth1 computes retrieval tasks)**.** *Fix any graph reasoning task among node count, edge count, edge existence, and node degree and any graph size $N$. Then, there exists a single-layer single-headed transformer $f \in \mathsf{Transformer}_{m,1,1}^N$ with embedding dimension $m = O(\log N)$ that solves the task on all graphs $G = (V, E)$ of size $|V| + |E| \le N - 1$ formatted as node/edge embedding sequences.*

*Proof.* We prove that a single head of self-attention with input and output MLPs can solve these retrieval and aggregation tasks in a parameter-efficient manner by first carefully designing a universal input MLP $\phi : \mathbb{R}^d \to \mathbb{R}^m$ for some $m = O(\log N)$ to produce embeddings that encode useful graph properties. Then, we define task-specific query, key, and value matrices $Q, K, V :\in \mathbb{R}^{m \times m}$ and output MLPs $\psi : \mathbb{R}^m \to \mathbb{R}$ that produce the correct answers. While we do not explicitly account for finite-precision computations, all of the operations utilized can be carefully implemented to respect $O(\log N)$-bit floating-point numerical representations using the technical approaches of [70, 71].

**Shared sinusoidal embedding MLP.** We denote the embedding output of the input MLP as

$$\phi(x_i) = (\texttt{isTask}_i, \texttt{isVertex}_i, \texttt{isEdge}_i, \phi'(x_i)) \in \mathbb{R}^m$$

for some $\phi'(x_i) : \mathbb{R}^d \to \mathbb{R}^{2m'}$ for some $m' = O(\log N)$ and $m = 2m' + 3$. For some fixed $a_1, \dots, a_{m'} \in [0, 1]$ to be determined, let

$$\phi'(x_i) = \begin{cases} \eta(x_i^1) & \text{if } \texttt{isVertex}_i = 1, \\ \xi(x_i^1, x_i^2) & \text{if } \texttt{isEdge}_i = 1, \\ \vec{0} & \text{otherwise}, \end{cases}$$

where $\eta$ is a sinusoidal embedding MLP for nodes with

$$\eta(v) = (\sin(2\pi a_1 v), \cos(2\pi a_1 v), \dots, \sin(2\pi a_{m'} v), \cos(2\pi a_{m'} v)) \in \mathbb{R}^{2m'}, \tag{8}$$

and $\xi$ is an edge embedding satisfying

$$\xi(u, v) = \frac{m'}{m' + \eta(u)^\mathsf{T} \eta(v)} (\eta(u) + \eta(v)).$$

We first show that the node embeddings are approximately orthogonal. By employing standard trigonometric identities, we obtain

$$\eta(u)^\mathsf{T} \eta(v) = \sum_{j=1}^{m'} \cos(2\pi a_j(u - v)),$$

and note that $\|\eta(v)\|_2^2 = m'$. We use a standard concentration argument to prove that $|\eta(u)^\mathsf{T} \eta(v)| \ll m'$ if $u \neq v$ with high probability.

**Claim 37** (Near-orthogonality of embeddings $\eta$). *There exist coefficients $a_1, \dots, a_{m'} \in [0, 1]$ that comprise the embedding $\eta : [n] \to \mathbb{R}^m$ of Equation (8) such that*

$$\left| \eta(u)^\mathsf{T} \eta(v) \right| \leq 2\sqrt{m' \log n}, \text{ if } u \neq v.$$

*Proof.* We employ the probabilistic method. Consider $m'$ iid random variables $\mathbf{a}_1, \dots, \mathbf{a}_{m'} \sim \text{Unif}([0, 1])$ and let $\boldsymbol{\eta}$ represent the respective node embedding. Fix some arbitrary $u, v \in [n]$ with $u \neq v$ and note that

$$\boldsymbol{\eta}(u)^\mathsf{T} \boldsymbol{\eta}(v) = \sum_{j=1}^{m'} \cos(2\pi \mathbf{a}_j(u - v)).$$

For any $j \in [m']$, the integrality of $u - v$ implies that

$$\mathbb{E}_{\mathbf{a}_j} \left[ \cos(2\pi \mathbf{a}_j(u - v)) \right] = 0.$$

Hoeffding's inequality provides the following bound:

$$\Pr\left[ \left| \boldsymbol{\eta}(u)^\mathsf{T} \boldsymbol{\eta}(v) \right| \geq 2\sqrt{m' \log n} \right] \leq \exp\left( -\frac{4m' \log n}{2m'} \right) \leq \frac{1}{n^2}.$$

By applying a union bound to all $\binom{n}{2}$ choices of $u$ and $v$, we have the following:

$$\Pr\left[ \forall u, v \in [n], \ u \neq v : \ \left| \boldsymbol{\eta}(u)^\mathsf{T} \boldsymbol{\eta}(v) \right| \geq 2\sqrt{m' \log n} \right] \leq \frac{n(n-1)}{2n^2} < 1.$$

Hence, there exists a satisfactory set of coefficients $a_1, \dots, a_{m'}$. $\qquad \square$

For some $\eta$ satisfying Claim 37, let $\chi = \max_{u, v \in [n], u \neq v} |\eta(u)^\mathsf{T} \eta(v)| \leq 2\sqrt{m' \log n}$. By taking $m' = O(\log n)$ be sufficiently large, we guarantee that $\chi \leq \frac{m'}{c}$ for any constant $c \geq 2$.

We then bound all relevant inner products between vertex and edge embeddings for sufficently large $m'$ (and sufficiently small $\chi$). We assume throughout that $u, u', v, v'$ are distinct and note that $\xi(u, v) = \xi(v, u)$ (and omit symmetric inequalities).

$$\|\eta(v)\|_2^2 = m'. \tag{9}$$

$$\left|\eta(u)^\mathsf{T}\eta(v)\right| \le \chi \le \frac{m'}{4}. \tag{10}$$

$$\left|\eta(u)^\mathsf{T}\xi(u,v) - m'\right| = \left|\frac{m'(m' + \eta(u)^\mathsf{T}\eta(v))}{m' + \eta(u)^\mathsf{T}\eta(v)} - m'\right| = 0. \tag{11}$$

$$\left|\eta(u)^\mathsf{T}\xi(u',v')\right| = \left|\frac{m'(\eta(u)^\mathsf{T}\eta(u') + \eta(u)^\mathsf{T}\eta(v'))}{m' + \eta(u')^\mathsf{T}\eta(v')}\right| \le \frac{2m'\chi}{m' - \chi} \le 4\chi \le \frac{m'}{4}. \tag{12}$$

$$\left|\|\xi(u,v)\|_2^2 - 2m'\right| = \left|\frac{m'^2 \cdot (2m' + 2\eta(u)^\mathsf{T}\eta(v)) - 2m'(m' + \eta(u)^\mathsf{T}\eta(v))^2}{(m' + \eta(u)^\mathsf{T}\eta(v))^2}\right|$$

$$\le \left|\frac{2m'\eta(u)^\mathsf{T}\eta(v)}{m' + \eta(u)^\mathsf{T}\eta(v)}\right| \le \frac{2m'\chi}{m' - \chi} \le 4\chi \le \frac{m'}{4}. \tag{13}$$

$$\left|\xi(u,v)^\mathsf{T}\xi(u,v') - m'\right| = \left|\frac{m'^2(m' + \eta(u)^\mathsf{T}\eta(v') + \eta(u)^\mathsf{T}\eta(v) + \eta(v)^\mathsf{T}\eta(v'))}{(m' + \eta(u)^\mathsf{T}\eta(v))(m' + \eta(u)^\mathsf{T}\eta(v'))} - m'\right|$$

$$= \left|\frac{m'^2\eta(v)^\mathsf{T}\eta(v') - m'\eta(u)^\mathsf{T}\eta(v)\eta(u)^\mathsf{T}\eta(v')}{(m' + \eta(u)^\mathsf{T}\eta(v))(m' + \eta(u)^\mathsf{T}\eta(v'))}\right|$$

$$\le \frac{m'^2\chi + m'\chi^2}{(m' - \chi)^2} \le 4\chi + 4\frac{\chi^2}{m'} \le \frac{m'}{4}. \tag{14}$$

$$\left|\xi(u,v)^\mathsf{T}\xi(u',v')\right| = \left|\frac{m'^2(\eta(u)^\mathsf{T}\eta(u') + \eta(u)^\mathsf{T}\eta(v') + \eta(v)^\mathsf{T}\eta(u') + \eta(v)^\mathsf{T}\eta(v'))}{(m' + \eta(u)^\mathsf{T}\eta(v))(m' + \eta(u')^\mathsf{T}\eta(v'))}\right|$$

$$\le \frac{4m'^2\chi}{(m' - \chi)^2} \le 16\chi \le \frac{m'}{4}. \tag{15}$$

Therefore, we conclude the following bounds:

$$\eta(u)^\mathsf{T}\eta(v), \ \eta(u)^\mathsf{T}\xi(u',v'), \ \xi(u,v)^\mathsf{T}\xi(u',v') \in \left[-\frac{m'}{4}, \frac{m'}{4}\right],$$

$$\|\eta(v)\|_2^2, \ \eta(u)^\mathsf{T}\xi(u,v), \ \xi(u,v)^\mathsf{T}\xi(u,v') \in \left[\frac{3m'}{4}, \frac{5m'}{4}\right],$$

$$\|\xi(u,v)\|_2^2 \in \left[\frac{7m'}{4}, \frac{9m'}{4}\right].$$

We now apply the above bounds to prove that each task can be solved by a single-headed transformer with weights $Q, K, V \in \mathbb{R}^{m \times m}$, output MLP $\psi : \mathbb{R}^m \to \mathbb{R}$, and shared input MLP $\phi$ discussed previously.

**Node count task.** If $P$ encodes the node count task, we specify weights matrices $Q, K, V$ in order to ensure that $Q^\mathsf{T}\phi(x_N) = m'e_1$[9]; $K^\mathsf{T}\phi(x_i) = (\texttt{isVertex}_i + \texttt{isTask}_i) \cdot e_1$; and $V^\mathsf{T}\phi(x_i) = \texttt{isTask}_i \cdot e_1$[10]. Then, the following is true of exponentiated key/query inner products and scalings products of value vectors:

$$\exp(\phi(x_N)^\mathsf{T}QK^\mathsf{T}\phi(x_i)) = \begin{cases} e^{m'} & \text{if } \texttt{isVertex}_i = 1 \text{ or } i = N, \\ 1 & \text{otherwise.} \end{cases}$$

$$\exp(\phi(x_N)^\mathsf{T}QK^\mathsf{T}\phi(x_i))V\phi(x_i) = \begin{cases} e^{m'} \cdot e_1 & \text{if } i = N, \\ \vec{0} & \text{otherwise.} \end{cases}$$

---

[9]Note that $Q^\mathsf{T}\phi(x_N)$ is the only relevant query embedding to the output $f(X)_N$; see Equation (7).

[10]While we do not specify the entries of $Q, K, V$, they are clearly linear transformations.

The output of the softmax is then

$$\text{softmax}\left(\phi(x_N)^\mathsf{T}QK^\mathsf{T}\phi(X)^\mathsf{T}\right)\phi(X)V = \frac{\sum_{i=1}^N \exp(\phi(x_N)^\mathsf{T}QK^\mathsf{T}\phi(x_i))V\phi(x_i)}{\sum_{i=1}^N \exp(\phi(x_N)^\mathsf{T}QK^\mathsf{T}\phi(x_i))}$$

$$= \frac{e^{m'}}{e^{m'}\cdot(1+|V|)+1\cdot(N-|V|-1)}\cdot e_1.$$

Let

$$y_N := \left(\text{softmax}\left(\phi(x_N)^\mathsf{T}QK^\mathsf{T}\phi(X)\right)\phi(X)V\right)_1 \in \mathbb{R}$$

denote the first coordinate of the softmax output. By taking $m' \geq \log(4N)$, we guarantee that

$$y_N \in \left[\frac{1}{1+|V|+N/e^{m'}}, \frac{1}{1+|V|}\right] \subseteq \left[\frac{1}{\frac{5}{4}+|V|}, \frac{1}{1+|V|}\right].$$

By letting the output MLP $\psi$ approximate the function $\psi(z) = \left\lfloor\frac{1}{z_1}-1\right\rfloor$ for $z_1 \in [1, N+2]$, we can ensure that $f(X)_N = |V|$.

**Edge count task.** We similarly let $Q^\mathsf{T}\phi(x_N) = m'e_1$; $K^\mathsf{T}\phi(x_i) = (\texttt{isEdge}_i + \texttt{isTask}_i)\cdot e_1$; and $V^\mathsf{T}\phi(x_i) = \texttt{isTask}_i\cdot e_1$. By an identical analysis of the attention matrix and $\psi$ construction, we ensure that $f(X)_N = |E|$.

**Edge existence task.** Under the node/edge encoding, we assume that the edge existence task is encoded as $x_N = (x_N^1, x_N^2, 0, 1, 1)$ for some $x_N^1, x_N^2 \in V$ and should return $f(X)_N = 1\left\{(x_N^1, x_N^2) \in E\right\}$. We choose our weight matrices to ensure that $Q^\mathsf{T}\phi(x_N) = \phi'(x_N) = \xi(x_N^1, x_N^2)$; $K^\mathsf{T}\phi(x_i) = \phi'(x_i)$, and $V^\mathsf{T}\phi(x_i) = 2(1-\texttt{isTask}_i)e_1$. By applying Claim 37 and letting $m' = O(\log N)$ to be sufficiently large, the following is true the query/key inner products:

$$\exp(\phi(x_N)^\mathsf{T}QK^\mathsf{T}\phi(x_i)) = \exp(\|\xi(x_N^1, x_N^1)\|_2^2) \geq e^{7m'/4} \quad \text{if } \{x_i^1, x_i^2\} = \{x_N^1, x_N^2\},$$

$$\exp(\phi(x_N)^\mathsf{T}QK^\mathsf{T}\phi(x_i)) \leq e^{5m'/4} \leq \frac{1}{8N}e^{7m'/4} \qquad\qquad \text{otherwise.}$$

We can therefore analyze the softmax output $y_N$ to obtain the following bound:

$$y_N \leq \frac{2\exp(\|\xi(x_N^1, x_N^1)\|_2^2)1\left\{(x_N^1, x_N^2) \in E\right\} + 2N\cdot\frac{1}{8N}e^{7m'/4}}{\exp(\|\xi(x_N^1, x_N^1)\|_2^2)(1+1\left\{(x_N^1, x_N^2) \in E\right\})}$$

$$\leq \frac{21\left\{(x_N^1, x_N^2) \in E\right\}+\frac{1}{4}}{1+1\left\{(x_N^1, x_N^2) \in E\right\}} \leq 1\left\{(x_N^1, x_N^2) \in E\right\}+\frac{1}{4},$$

$$y_N \geq \frac{2\exp(\|\xi(x_N^1, x_N^1)\|_2^2)1\left\{(x_N^1, x_N^2) \in E\right\}}{\exp(\|\xi(x_N^1, x_N^1)\|_2^2)(1+1\left\{(x_N^1, x_N^2) \in E\right\}) + N\cdot\frac{1}{8N}e^{7m'/4}}$$

$$\geq \frac{21\left\{(x_N^1, x_N^2) \in E\right\}}{1+1\left\{(x_N^1, x_N^2) \in E\right\}+\frac{1}{8}} \geq 1\left\{(x_N^1, x_N^2) \in E\right\}-\frac{1}{4}.$$

Hence,

$$y_N \in \begin{cases}\left[-\frac{1}{4}, \frac{1}{4}\right] & \text{if } (x_N^1, x_N^2) \notin E, \\ \left[\frac{3}{4}, \frac{5}{4}\right] & \text{if } (x_N^1, x_N^2) \in E.\end{cases}$$

Therefore, it suffices to design a threshold output MLP $\psi$ that satisfies

$$\psi(z) = \begin{cases}1 & z_1 \geq \frac{3}{4}, \\ 0 & z_1 \leq \frac{1}{4},\end{cases}$$

in order to distinguish between the two possible output ranges. This can be easily constructed by taking a linear combination of two ReLU neurons.

**Node degree task.** We assume that the task is encoded as $x_N = (x_N^1, 0, 1, 0, 1)$ and should return $f(X)_N = \deg(x_N^1) := \left| \{(x_N^1, v) \in E\} \right|$. We use weight matrices with $Q^\mathsf{T}\phi(x_N) = \phi'(x_N) = \eta(x_N)$; $K^\mathsf{T}\phi(x_i) = \phi'(x_i)$; and $V^\mathsf{T}\phi(x_i) = \mathtt{isTask}_i$. This ensure that the following is true about the query/key inner products for sufficiently large $m'$:

$$\exp(\phi(x_N)^\mathsf{T} QK^\mathsf{T}\phi(x_i)) = e^{m'} \qquad\qquad \text{if } i = N \text{ or } x_N^1 \in \{x_i^1, x_i^2\},$$

$$\exp(\phi(x_N)^\mathsf{T} QK^\mathsf{T}\phi(x_i)) \le e^{m'/4} \le \frac{1}{4N}e^{m'} \qquad\qquad \text{otherwise.}$$

We similarly ensure that the following holds about the softmax output $y_N$.

$$y_N \le \frac{e^{m'}}{e^{m'}\cdot(\deg(x_N^1)+2)} = \frac{1}{\deg(x_N^1)+2},$$

$$y_N \ge \frac{e^{m'}}{(\deg(x_N^1)+2)e^{m'} + N\cdot\frac{1}{4N}e^{m'}}$$

$$\ge \frac{1}{\deg(x_N^1)+\frac{9}{4}}.$$

We then use the similar approach employed in the node count task to choose an output MLP that approximately computes $\phi(z) = \left\lfloor \frac{1}{z_1} - 2 \right\rfloor$.

We thus conclude that there exist single-layer transformers that solve all of the described tasks. $\square$

## C.2 Negative results for single-layer transformers

**Theorem 38** (Formal version of Theorem 6; Depth1 cannot compute search or retrieval tasks). *Fix any graph reasoning task among graph connectivity, shortest path, and cycle detection. Any single-layer transformer $f \in \mathsf{Transformer}_{m,H,1}^N$ with $O(\log N)$-bit precision that solves the task on all graphs $G = (V, E)$ of size $|V| + |E| \le N - 1$ formatted as node/edge embedding sequences has width satisfying $mH = \Omega(N/\log N)$.*

Our negative results generalizes and applies the approach of [70] to prove negative results for single-layer transformers by communication complexity reductions. The bounds hinge on the following fundamental fact about the hardness of a two-player game where two agents jointly attempt to compute a set disjointness quantity that depends on both of their inputs with bounded communication.

**Fact 1** (Disjointness communication complexity lower bound [85]). *Suppose Alice and Bob are given inputs $a, b \in \{0,1\}^r$ and wish to jointly compute $\mathrm{DISJ}(a, b) = \max_i a_i b_i$ by alternately sending single-bit messages to one another. Any deterministic protocol that computes $\mathrm{DISJ}(a, b)$ requires at least $r$ rounds of communication, or $r$ bits of information.*

We first generalize Theorem 7 of [70] to show that no transformer can efficiently solve an embedded set disjointness problem without having a width that scales linearly in $r$. We prove the theorem and later apply to it prove negative results about relevant graph reasoning tasks. To demonstrate that the graph reasoning tasks do not require pathological input graphs $G$ to be hard, we exhibit particular input graph instances with constant graph diameter or constant degree where the task cannot be efficiently solved.

**Lemma 39** (Generic Depth1 communication complexity negative result). *For some sequence length, fix two disjoint subsets $A, B \subset [N - 1]$, and consider a single-layer transformer $f \in \mathsf{Transformer}_{m,H,1}^N$ with $O(\log N)$-bit precision that solves set disjointness, i.e. $f(X)_N = \mathrm{DISJ}(a, b)$ for any input $X$ where $X_A$ is a function of Alice's input $a \in \{0,1\}^r$, $X_B$ is a function of Bob's input $b \in \{0,1\}^r$, and $X_{[N]\setminus(A\cup B)}$ is fixed regardless of $a$ and $b$. Then, $f$ has width satisfying $mH = \Omega(r/\log N)$.*

*Proof.* Consider any transformer $f$ that solves $\mathrm{DISJ}$ as specified in the theorem statement. We show that this implies the existence of a $O(mH \log N)$-round communication protocol that solves $\mathrm{DISJ}(a, b)$ for any inputs $a, b \in \{0,1\}^r$. An application of Fact 1 immediately proves the theorem statement.

If such a transformer $f$ exists, then the following is true for some $\phi, \psi$, and $Q, K, V$:

$$\text{DISJ}(a,b) = \psi\left(\sum_{h=1}^{H} \text{softmax}\left(\phi(x_N)^\mathsf{T} Q_h K_h^\mathsf{T} \phi(X)\right) \phi(X) V_h\right)$$

$$= \psi\left(\sum_{h=1}^{H} \frac{Z_{h,A}\exp(L_{h,A}) + Z_{h,B}\exp(L_{h,B}) + Z_{h,[N]\setminus(A\cup B)}\exp(L_{h,[N]\setminus(A\cup B)})}{\exp(L_{h,A}) + \exp(L_{h,B}) + \exp(L_{h,[N]\setminus(A\cup B)})}\right),$$

for partial softmax and normalization terms[11] defined as follows for $S \subset [N]$ and $h \in [H]$:

$$Z_{h,S} = \frac{\sum_{i\in S}\exp(\phi(x_N)^\mathsf{T} Q_h K_h^\mathsf{T}\phi(x_i))V_h\phi(x_i)}{\sum_{i\in S}\exp(\phi(x_N)^\mathsf{T} Q_h K_h^\mathsf{T}\phi(x_i))} \in \mathbb{R}^m,$$

$$L_{h,S} = \log\left(\sum_{i\in S}\exp(\phi(x_N)^\mathsf{T} Q_h K_h^\mathsf{T}\phi(x_i))\right) \in \mathbb{R}_+.$$

The definition of input instances $X$ implies that Alice can compute $Z_{h,A}$ and $L_{h,A}$ as a function of her input $a$; Bob can similarly compute $Z_{h,B}$ and $L_{h,B}$ from $b$; and $Z_{h,[N]\setminus(A\cup B)}$, $L_{h,[N]\setminus(A\cup B)}$, and all implementation details of $f$ are known by all players. Therefore, Bob can compute $\text{DISJ}(a,b)$ by an $O(mH\log N)$-round protocol where Alice sends him $(Z_{h,A}, L_{h,A})$ bit-by-bit. $\square$

It remains to apply Lemma 39 to each graph reasoning task by defining graph instance encodings $X$ that encode $\text{DISJ}(a,b)$ in the solution to the task for some $r = \Theta(N)$.

*Proof of Theorem 38.* For each task, we provide a pair of "hard instances:" one with constant degree and one with constant diameter, in order to obey different notions of graph simplicity.

**Graph connectivity task.** For both instances, on any disjointness input $a, b \in \{0,1\}^r$, we define a graph $G = (V, E)$ with $|V| + |E| = O(r)$ whose edges encode the input, i.e. $E = E(a,b)$. We define three fixed disjoint sets of potential edges $\bar{E}_A, \bar{E}_B, \bar{E}_*$ with $|\bar{E}_A| + |\bar{E}_B| + |\bar{E}_*| = O(r)$ such that $E = E_A(a) \cup E_B(b) \cup \bar{E}_*$, where $E_A(a) \subset \bar{E}_A$ is a function of only Alice's input $a$ and $E_B(b) \subset \bar{E}_B$ is a function of Bob's $b$.

We define an enumeration of all vertices in $V$ and potential edges in $\bar{E}_A, \bar{E}_B, \bar{E}_*$. We first fix their sizes $|V|, |\bar{E}_A|, |\bar{E}_B|, |\bar{E}_*|$ and then index the vertices and edges in the order $V, \bar{E}_A, \bar{E}_B, \bar{E}_*$. That is,

$$V = \{1, \ldots, |V|\},$$
$$\bar{E}_* = \{(u_i, v_i) : i \in S_*\}, \text{ for } S_* = \{|V|+1, \ldots, |V|+|\bar{E}_*|\},$$
$$\bar{E}_A = \{(u_i, v_i) : i \in A\}, \text{ for } A = \{|V|+|\bar{E}_*|+1, \ldots, |V|+|\bar{E}_*|+|\bar{E}_A|\},$$
$$\bar{E}_B = \{(u_i, v_i) : i \in B\}, \text{ for } B = \{|V|+|\bar{E}_*|+|\bar{E}_A|+1, \ldots, |V|+|\bar{E}_*|+|\bar{E}_A|+|\bar{E}_B|\}.$$

This implies that the following transformer input $X \in R^{N\times d}$ for $N = |V| + |S_*| + |A| + |B| + 1$ is a valid node/edge encoding such that inputs $X_A$ is a function of $a$, $X_B$ is a function of $b$, and $X_{[N]\setminus(A\cup B)}$ is constant:

- For $i \in V$, $x_i = (i, 0, 1, 0, 0)$ encodes a fixed vertex.

- For $i \in S_*$, $x_i = (u_i, v_i, 0, 1, 0)$ encodes a fixed edge.

- For $i \in A$, $x_i$ represents edge $(u_i, v_i)$ if it exists:

$$x_i = \begin{cases} (u_i, v_i, 0, 1, 0) & \text{if } (u_i, v_i) \in E_A(a), \\ (0, 0, 0, 0, 0) & \text{otherwise.} \end{cases}$$

---

[11]These terms are designed in order to facilitate computation with $\log(N)$-bit floating-point numbers. Note that each $Z_{h,S}$ is a convex combination of value vectors $V_h^\mathsf{T}\phi(x_i)$, which means that $Z_{h,S}$ requires no more bits of precision to accurately approximate than $V_h^\mathsf{T}\phi(x_i)$. Similarly, each $L_{h,S}$ is within an $O(\log N)$ additive factor of $\max_{i\in S}\phi(x_N)^\mathsf{T} Q_h K_h^\mathsf{T}\phi(x_i)$. Finally, the output of the softmax term for head $h$ is a convex combination of $Z_{h,A}, Z_{h,B}$, and $Z_{h,[N]\setminus(A\cup B)}$, which makes accurate computation of each output possible without requiring greater precision.

- For $i \in B$, $x_i$ represents edge $(u_i, v_i)$ if it exists:

$$x_i = \begin{cases} (u_i, v_i, 0, 1, 0) & \text{if } (u_i, v_i) \in E_B(b), \\ (0, 0, 0, 0, 0) & \text{otherwise.} \end{cases}$$

- $x_N = (u, v, 0, 1, 1)$ encodes the task token for some fixed $(u, v) \in V^2$.

*Constant-diameter instance:* We define a graph $G$ of diameter at most 8 that encodes an disjointness instance $a, b \in \{0, 1\}^r$ in an instance of connectivity.

- Let $|V| = 3r+2$, and let nodes 1 and $3r+2$ denote the "source" and "sink" node respectively. That is, $x_N = (1, 3r + 2, 0, 1, 1)$.

- Edges between the source node 1 and nodes $\{2, \ldots, r+1\}$ and between the end node $3r+2$ and $\{2r+2, \ldots, 3r+1\}$ always are included. That is,

$$E_* = \{(1, i) : i \in \{2, \ldots, r+1\}\} \cup \{(3r+2, i) : i \in \{2r+2, \ldots, 3r+1\}\}.$$

- Alice's input is encoded as edges $(i+1, i+r+1)$ for $i \in [r]$. That is,

$$E_A(a) = \{(i+1, i+r+1) : a_i = 1\} \subset \bar{E}_A = \{(i+1, i+r+1) : i \in [r]\}.$$

- Bob's inputs are similarly encoded as $(i+r+1, i+2r)$:

$$E_B(b) = \{(i+r+1, i+2r+1) : b_i = 1\} \subset \bar{E}_B = \{(i+r+1, i+2r+1) : i \in [r]\}.$$

We visualize this construction of $G$ in Figure 5.

There exists a path between node 1 and node $3r + 2$ if and only if there exists some $i \in [r]$ such that $(i+1, i+r+1), (i+r+1, i+2r+1) \in E$, which corresponds to $a_i = b_i = 1$. Thus, the graph $G$ is connected if and only if $\text{DISJ}(a, b) = 1$. Any transformer $f$ that computes

$$f(X)_N = 1 \{1 \text{ and } 3r + 2 \text{ are connected in} G\}$$

also solves disjointness. Since $N = |V| + |S_*| + |A| + |B| + 1 = \Theta(r)$, we conclude the proof of hardness of solving graph connectivity on this instance by applying Lemma 39.

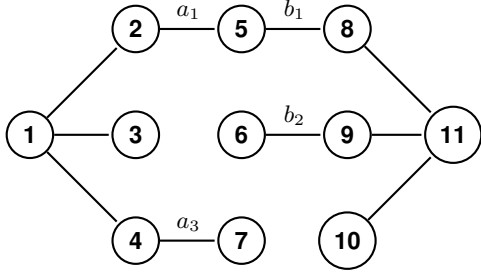

Figure 5: The constant diameter graph construction for $r = 3$, $A = (1, 0, 1)$ and $B = (1, 1, 0)$. The source 1 and sink 11 are connected which is equivalent to $\text{DISJ}(A, B) = 1$ by construction.

*Constant-degree instance:* We can define a degree-3 graph $G$ that similarly encodes a disjointness instance $a, b$. To do so, we modify the previous construction by replacing the fixed edges $E_*$ with two binary trees, each of depth $O(\log r)$, between the source and end nodes as roots and Alice and Bob's nodes incident to $\bar{E}_A$ and $\bar{E}_B$ respectively as leaves. The remainder of the construction—including the encoding of $E_A(a)$ and $E_B(b)$ and the connectivity analysis—is identical. Since a binary tree with $r$ leaves has $O(r)$ nodes and edges, the graph $G$ can be similarly encoded as a length-$N$ transformer input for $N = O(r)$. See Figure 6 for a visualization.

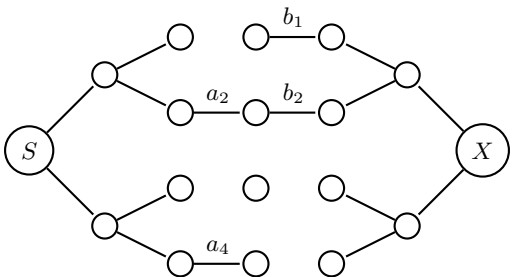

Figure 6: The degree 3 graph construction for $r = 4$, $A = (0, 1, 0, 1)$, $B = (1, 1, 0, 0)$. The source node $S$ and sink node $X$ are connected, which is equivalent to $\text{DISJ}(A, B) = 1$ by construction.

**Shortest path task.** We can modify the graph connectivity constructions $G$ to create a decision variant of shortest path, rather than graph connectivity. Let $D(G)$ be the length of a path between the source and sink node in each connectivity construction, and note that $D(G) = 4$ for the constant-diameter instance and $D(G) = O(\log r)$ for constant-degree. We design a new graph $G'$ of size $O(r)$ then appends a path of $D(G) + 1$ vertices to $G$. Then, the shortest path between the source and sink node is $D(G)$ if they are connected and $D(G) + 1$ if not.

If there exists a transformer $f'$ solves shortest path on graphs of the form $G'$, then there exists a minor modification $f$ that solves graph connectivity on graph $G$.

**Cycle check task.** We again modify the graph connectivity constructions. We produce $G'$ by adding a single edge to $G$ between the source node 1 and the sink node $3r + 2$, which ensures that $G'$ has a cycle if and only if $G$ is connected between the source and sink nodes. Therefore, a transformer $f'$ can only solve check check if there exists a transformer $f$ that solves connectivity. $\square$

# D   Representational gaps between transformers and GNNs

Our experiments in Section 4 highlight the differences in graph reasoning abilities between vanilla transformers with a naive input graph tokenization and graph neural networks (GNNs). These distinctions can be understood as the consequences of model capacity or inductive bias between transformers and GNNs. This appendix contrasts the novel analysis of the capacities of vanilla transformers in Section 3 with previously established limitations on GNNs. We present two theoretical tools for deriving negative results on GNN capabilities: CONGEST distributed computing model and the Weisfeiler-Leman isomorphism test (WL-test). We discuss the results implied by these frameworks and contrast them with transformer capabilities and limitations.

## D.1   Limitations of GNNs via CONGEST analogy

The bidirectional relationship between transformers and the massively parallel computation (MPC) distributed computing model of [71] was partially inspired by a similar analogy between GNNs and the CONGEST model by [51].

The MPC and CONGEST distributed computing protocols differ in their models of how messages are passed between machines. While MPC protocols permit any machine to send a message to any other machine subject to capacity constraints, CONGEST protocols operate on a graph topology that restricts machines to send messages exclusively to and from neighbors. As a consequence, two nodes in a CONGEST graph that are separated by a path of length $p$ cannot share information with one another without $\Omega(p)$ rounds of communication; no similar notion of "long-range communication" exists for MPC.

All message-passing GNNs where each node initially knows only its own features can be simulated by a CONGEST protocol whose rounds scales linearly in the depth of the GNN [51]. This reduction implies a lower bound on solving several parallelizable and search tasks discussed in Appendix A.2.

**Theorem 40.** *Any graph neural network $f$ with width (message size) $m$ and depth $L$ that computes any of the following tasks on $n$-node graphs has the following size lower bounds:*

- *Subgraph connectivity, minimum spanning forest, minumum cut, shortest path: $L\sqrt{m} = \tilde{\Omega}(\sqrt{n})$.*
- *Cycle detection, diameter computation: $Lm = \tilde{\Omega}(n)$.*

*If $L = O(\log N)$, then for all tasks, $m = \tilde{\Omega}(n)$.*

In contrast, the quantitative bounds in Section 3 establish sharp trade-offs between transformers and GNNs for parallelizable tasks and suggest a possible equivalence for search tasks.

All parallelizable tasks—including (subgraph) connectivity, minimum spanning forest, minimum cut, and cycle detection—can be solved by transformers of depth $L = O(\log n)$ and width $m = O(n^\epsilon)$ for any constant $\epsilon \in (0, 1)$ due to Theorem 2. In contrast, Theorem 40 requires that a similar-depth GNNs have width $m = \tilde{\Omega}(n)$, and a GNN of comparable width requires depth $L = \tilde{\Omega}(n^{1-\epsilon})$.

On the other hand, search tasks, such as shortest path and diamter, are only guaranteed to be solvable by transformers of depth $O(\log n)$ and width $O(n^{1+\epsilon})$ (for graphs with $|E| = n^2$) by Theorem 4. This statement compares to the GNN negative results of Theorem 40.

### D.2 Limitations of GNNs via the Weisfeiler-Leman test

A relationship between GNNs and the Weisfeiler-Leman heuristic graph isomorphism test [83] (WL test) further establishes representational limitations of message-passing GNNs. This connection and the rich literature surrounding it is presented in greater detail by [59].

The 1-WL test is a permutation-invariant test for predicting whether two candidate graphs are isomorphic that works by first labeling each node with the empty set $\emptyset$ and then repeatedly replacing each label with a multiset of its neighbors' labels. A hierarchy of WL test variants exists where the $k$-WL test maintains a label for each $k$-tuple of vertices. The failure models of these heuristic solutions are well-understood; critically, the 1-WL test cannot distinguish between connected and disconnected graphs.

The abilities of message-passing GNNs without unique node identifiers to determine whether two graphs are isomorphic are limited by the graphs distinguishable by the 1-WL test [84, 59]. As an immediate consequence, such GNNs cannot solve graph connectivity *at all*, unlike transformers, which can do so with a logarithmic-depth parameter-efficient representation. The relationship is bidirectional; message-passing GNNs (and transformers as well) admit an efficient approximation of the 1-WL test.

The effectiveness of these bounds is tempered by the assumption that no node includes identifying features, which is easily overcome by standard GNN models. The analogy is further limited by the fact that graph embeddings of transformers must have some (possibly arbitrary) node identifier as input in order to tokenize a graph using the node/edge encoding without losing the ability to associate each node with its incident edges. However, the juxtaposition of the 1-WL and CONGEST-based limitations on the abilities of GNNs to solve connectivity-based tasks suggests a fundamental gap in capabilities between models that is apparent in multiple theoretical lenses.

## E Experimental Details

### E.1 Datasets

We evaluate our model on the diverse graph reasoning tasks presented in GraphQA [24]. We used the public code of the dataset available at https://github.com/google-research/google-research/tree/master/graphqa. The code to generate the datasets is licensed under the Apache License, Version 2.0. The tasks in the dataset range in difficulty and encompass the following categories:

- **Graph-level:** node counting (counting the number of nodes in a graph), edge counting (counting the number of edges in a graph), cycle check (determining whether a graph contains a cycle), and triangle counting (counting the number of triangles in a graph).
- **Node-level:** node degree (calculating the degree of a given node in a graph).

- **Edge-level:** connectivity (finding if there is a path from one node to another), edge existence (whether a given edge exists in a graph, and shortest path (finding the length of the shortest path from one node to another).

The graphs used in the experiments in this paper and the corresponding graph reasoning tasks are taken from [24]. There are $1,000$ graphs in the original train set, $500$ graphs in the dev set, and $500$ graphs in the test set. The graphs are generated randomly using Erdős-Rényi (ER) random graph model [23]. Graph size ranges from 5 to 20 nodes.

**Train set statistics.** Average number of nodes: 11.90; average number of edges: 37.01; average node degree: 5.43.

**Test set statistics.** Average number of nodes: 12.37; average number of edges: 39.79; average node degree: 5.70.

While random instances of graph reasoning tasks provide a valuable assessment of the task complexity on realistic graphs, they do not necessarily reflect the "worst case" graph inputs that convey negative results like Theorem 6 and Theorem 3. For example, the reduction that establishes that cycle check is "as hard as" graph connectivity and the consequential logarithmic-depth hardness results hinge on the consideration of graph instances with $n$ nodes and polynomial cycle length. However, as witnessed by Figure 7, the shortest cycles observed in 1000 instances of GraphQA cycle check is almost always of length three, and only 3.2% of instances are larger. As a consequence, identifying the existence of a cycle on the GraphQA dataset is inherently local, which is reflected by a strong performance by heuristic-based GNN solutions

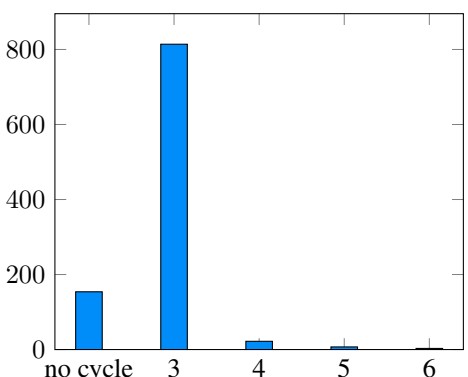

Figure 7: Histogram of minimum cycle lengths for cycle check instances.

(Table 2)—despite the fact that efficient GNNs for *worst-case* cycle check do not exist (Theorem 40).

For our experiments on the effect of the scale of the number of training data points in the final results we obtain, we use the open-source code of GraphQA available to generate a larger training dataset of 100K examples. We follow the original instructions and parameters to create this larger training dataset.

### E.2 Implementation Details

**Model Hyperparameters.** We fixed the number of iterations as 1,000,000 and train standard decoder-only transformers with $L = 12$ layers, $m = 768$ embedding dimension, $H = 12$ heads, learning rate $5 \cdot 10^{-4}$, and dropout 0.1. These models have an approximate parameter count of 60,000,000.

We used random search [9] over the following set of hyperparameters to select a universal architecture for all tasks: The range provided for the learning rate and dropout rate are $[10^{-4}, 10^{-1}]$ and $[0, 0.5]$. The number of layers $L$ and embedding dimension $m$ is selected from $L \in \{4, 6, 8, 10, 12, 14, 16\}$ and $m \in \{192, 384, 576, 768, 960, 1152, 1344, 1536\}$. We employed the GLU [73] activation as a non-linearity.

**Model Selection.** We implemented our model in JAX [26] and used AdamW [43, 50] as the optimizer. Optimal hyperparameters for each task and model were determined by training on the GraphQA$_{Train}$ dataset and evaluating performance on the GraphQA$_{Dev}$ dataset. The results presented in the paper are based on the held-out GraphQA$_{Test}$ dataset.

**Hardware Acceleration.** All experiments were conducted using Google's TPUv3 and TPUv5e accelerators [35].

### E.3 Baseline Results

To rigorously evaluate the performance of transformers on graph reasoning tasks, we compare them against three established categories of baselines:

1. **Prompting-based methods.** These methods provide the LLM with a textual descriptions of the graph and question within the prompt. We consider the following variations and copy the results from the original papers:
   - ZERO-SHOT. In this approach, the model is given a task description and immediately asked to produce the desired output. No additional examples or demonstrations are provided.
   - FEW-SHOT. This approach provides the model with a few examples of the task and their desired outputs [12]. Unlike traditional training, these examples are included directly in the prompt, allowing the model to learn and adapt during the inference.
   - CoT. Chain-of-thought (CoT) prompting [82] provides examples each showing step-by-step reasoning, teaching the LLM to generate its own thought processes for tackling new tasks.
   - ZERO-COT. Zero-shot CoT [45] builds upon Chain-of-Thought (CoT) prompting by eliminating the need for training examples. The LLM generates its own step-by-step reasoning process using a simple trigger phrase like "Let's think step by step".
   - COT-BAG. BAG prompting [80] extends COT to improve the performance of LLMs on graph-related tasks by appending "Let's construct a graph with the nodes and edges first" to the prompt.

2. **Graph-based methods.** These models are specifically designed to process graphs as input and are trained task-specific. They leverage the connections between nodes to learn patterns and make predictions, making them ideal for tasks where a graph is involved. We use GCN [44], MPNN [28], and GIN [84] from this category. GraphToken [63] is a GNN-based model that processes the graph and feed the output of the GNN as soft-tokens to an LLM. We consider both GNN models with and without identity tokenizations, as the former is known to be theoretically limited by WL tests, while the latter poses a "fairer" comparison with the positional encodings of transformers.

3. **Transformer models (Ours).** The last class of model are task-specific vanilla transformer models [77]. The *60M transformer-1K* model is the one described above trained on $1,000$ training examples from the GraphQA training set. To investigate the impact of training data scale, we generated a larger dataset containing $100,000$ examples, ensuring the same distribution as the original training set by using the official GraphQA code and trained *60M transformer-100K* on that. The *11B transformer (FT)-1K* is a vanilla transformer model that is started with a pre-trained checkpoint of T5 [66] and is fine-tuned on the 1K training dataset. We also include two fine-tuned PaLM 2 [5] transformers of size XXS and XS. Similar to prompting baselines, this model receives a textual description of the graph as input to leverage its textual reasoning capabilities.

The results for ZERO-SHOT, ZERO-COT, FEW-SHOT, COT, and COT-BAG are taken from Fatemi et al. [24]. Results for SOFT-PROMPT and GraphToken are sourced from Perozzi et al. [63].

We independently evaluated GCN, MPNN, and GIN models on these tasks. We used the original architectures proposed in their respective papers and performed hyperparameter tuning on the GraphQA$_{Dev}$ dataset.

### E.4 Further Experimental results

Table 4 presents a comprehensive comparison of the graph reasoning capabilities across various baseline models and our proposed transformer architectures. The results highlight several key findings, which we summarize below:

**Transformers Exhibit Strong Performance on Graph-based Reasoning Problems.** While transformers are not explicitly designed for graph reasoning tasks like graph-based models, they demonstrate surprisingly strong performance in this domain. The results of this study indicate that transformers, despite their versatility as a general architecture, can often match or even surpass specialized graph models on a variety of graph reasoning benchmarks.

**Transformers Excel at Retrieval Tasks.** As proved in Theorem 36, retrieval tasks can be solved by transformers. The obtained results confirm that such tasks are relatively easy for transformers as they

| | | Retrieval Tasks | | | | Parallelizable Tasks | | Search Tasks | Subgraph Counting |
|---|---|---|---|---|---|---|---|---|---|
| Method | Samples | Node count | Edge count | Edge existence | Node degree | Connectivity | Cycle check | Shortest path | Triangle counting |
| ZERO-SHOT [24] | | 21.7 | 12.4 | 44.5 | 14.0 | 84.9 | 76.0 | 11.5 | 1.5 |
| ZERO-COT [24] | | 14.6 | 9.4 | 33.5 | 10.4 | 73.5 | 32.3 | 33.6 | 12.7 |
| FEW-SHOT [24] | | 25.3 | 12.0 | 36.8 | 17.4 | 79.4 | 37.4 | 22.7 | 3.0 |
| COT [24] | | 27.6 | 12.8 | 42.8 | 29.2 | 45.2 | 58.0 | 38.6 | 8.1 |
| COT-BAG [24] | | 26.9 | 12.5 | 37.3 | 28.0 | 45.2 | 52.1 | 40.4 | 8.1 |
| GCN (equivariant) [44] | 1K | 6.4 | 1.2 | 47.0 | 9.8 | 83.8 | 83.2 | 50.2 | 4.0 |
| GCN (identity tokenization) [44] | 1K | | | | 8.8 | 83.8 | 83.2 | | |
| GCN (identity tokenization) [44] | 100K | | | | 42.2 | 93.2 | 91.2 | | |
| MPNN (equivariant) [28] | 1K | 19.4 | 16.2 | 69.2 | 99.4 | 94.0 | 99.0 | 66.8 | 30.6 |
| MPNN (identity tokenization) [28] | 1K | | | | 38.8 | 94.0 | 98.6 | | |
| MPNN (identity tokenization) [28] | 100K | | | | 100.0 | 93.8 | 99.8 | | |
| GIN (equivariant) [84] | 1K | 71.2 | 4.4 | 71.2 | 36.2 | 93.8 | 98.8 | 54.0 | 30.4 |
| GIN (identity tokenization) [84] | 1K | | | | 37.8 | 93.4 | 97.6 | | |
| GIN (identity tokenization) [84] | 100K | | | | 100.0 | 94.0 | 98.6 | | |
| GraphToken [63] | 1K | 99.6 | 42.6 | 73.8 | 96.2 | 93.2 | 95.6 | 63.8 | 34.8 |
| 60M transformer | 1K | 100.0 | 100.0 | 67.6 | 31.5 | 92.9 | 97.1 | 57.4 | 33.4 |
| 60M transformer | 100K | 100.0 | 100.0 | 96.1 | 91.7 | 98.0 | 98.0 | 97.2 | 40.5 |
| XXS transformer (FT) | 1K | 100.0 | 70.6 | 73.0 | 31.0 | 93.6 | 98.0 | 60.4 | 29.0 |
| XS transformer (FT) | 1K | 100.0 | 73.2 | 98.6 | 50.6 | 96.6 | 96.8 | 60.0 | 28.6 |
| 12B transformer (FT) | 1K | 100.0 | 45.0 | 100.0 | 68.8 | 98.4 | 98.0 | 92.8 | 26.0 |

Table 4: Comparison of various methods in different categories on graph reasoning tasks of GraphQA. Here, we categorize the tasks using the taxonomy proposed in Section 3.

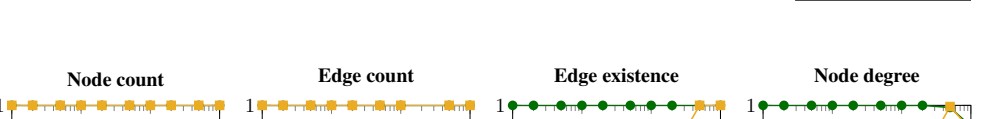

Figure 8: Comparison of train and test scaling on all tasks.

obtained the full accuracy on most of such tasks. One exception here is the node degree task that GNNs outperform transformers but still transformers perform relatively well. We discuss why GNNs outperform transformers well for this task.

**Larger Transformers Excel at Solving Search Tasks.** As discussed in Theorem 4, transformers are effective for search tasks, albeit requiring a larger number of parameters compared to retrieval tasks. This is empirically evident in the comparison between Transformer-1K and Transformer-1K (pretrained). It's worth noting that the pretrained transformer used here has 11 billion parameters, a significant increase from the 1 million parameters in Transformer-1K.

**Transformers Excel at Capturing Global Patterns.** An interesting observation here is the performance gap between Transformers and GNNs across tasks with varying emphasis on local versus global graph structure. Notably:

| Method | Samples | Retrieval Tasks | | | | Parallelizable Tasks | | Search Tasks | Subgraph Counting |
|---|---|---|---|---|---|---|---|---|---|
| | | Node count | Edge count | Edge existence | Node degree | Connectivity | Cycle check | Shortest path | Triangle counting |
| 60M transformer | 1K | 100.0 (0.00) | 100.0 (0.00) | 67.6 (0.94) | 31.2 (1.49) | 92.7 (1.01) | 96.2 (0.64) | 56.7 (1.60) | 26.6 (0.73) |
| 60M transformer | 100K | 100.0 (0.00) | 100.0 (0.00) | 100.0 (0.00) | 93.3 (11.80) | 97.4 (0.33) | 99.0 (0.14) | 97.7 (0.66) | 39.1 (0.72) |

Table 5: Mean accuracy (and standard deviation) of trained 60M-parameter transformers over five random seeds.

1. **Local Structure:** The node degree task, which relies heavily on local node information, is best handled by MPNN, a GNN-based model.

2. **Global Structure:** In contrast, tasks like connectivity, triangle counting and shortest path, which require understanding global graph patterns, are dominated by Transformer models. Notably, vanilla Transformers achieve a remarkable 45% relative improvement over even much larger LLMs augmented with GNN-generated soft prompts (GraphToken) on the shortest path task. This showcases the exceptional ability of Transformers to capture long-range dependencies, a critical factor in understanding global graph structure.

### E.4.1 Sample complexity ablations

To develop a more comprehensive understanding of graph reasoning tasks learnability by small transformers, we train a variety of transformers for 1,000,000 steps on each task on a range of sample sizes. In Figure 8, we demonstrate how the model performance improves as a function of sample complexity. By doing so, we witness the relative hardness of each task in terms of the marginal benefit of new samples.

- Edge count and node count are "easy" retrieval tasks that can be solved perfectly with as few as 100 samples.

- Edge existence attains near-perfect train and test classification accuracy as the number of training samples approaches 100,000.

- Connectivity, shortest path, and node degree demonstrate a sharp improvement in evaluation error as a function of the sample size. These models perfectly fit the training set in most sample size regimes, but yield a closer correspondence between training and testing error when trained on 100,000 samples.

- Cycle check and triangle count have persistent gaps between training and testing error and overfit even in the large sample setting.

### E.4.2 Experimental stability

While we lack computational resources to train all models multiple times to obtain error bars, we repeated the full experiments on 60M-parameter transformers with 1000 and 100,000 training samples on each task with five different random seeds in order to estimate the standard deviation of the resulting task accuracy and displayed those results in Table 5. With the exception of the node degree experiments trained on 100,000 samples (which included a single far-outlier), all tasks and sample regimes had no more than two percentage points of standard deviation. Critically, this implies the robustness of the superiority of trained transformers over GNNs and prompting methods on "intrinsically global" tasks, such as graph connectivity.

