# OpenReview forum: "Understanding Transformer Reasoning Capabilities via Graph Algorithms"
_NeurIPS.cc/2024/Conference — NeurIPS 2024 poster_

### Official Review · Reviewer_43vz · 2024-07-12

**Soundness:** 3
**Presentation:** 3
**Contribution:** 2
**Rating:** 6
**Confidence:** 3

**Summary:**

This paper investigates the algorithmic reasoning capabilities of transformers on graph problems and introduces a novel hierarchy that categorizes tasks into solvable classes under different scaling regimes of transformers. In addition, the authors perform experiments on GraphQA to validate their theoretical analysis.

**Strengths:**

1. **Theoretical Aspect**: It provides a novel hierarchy that categorizes graph problems based on the scale of transformers required to solve these problems, providing insights into their algorithmic reasoning capabilities.
2. **Empirical Validation**: This paper supports its theoretical analysis with some experiments on the GraphQA benchmark.

**Weaknesses:**

1. **Limited Generalizability**: The theoretical results are based on specific assumptions, parameter scaling regimes, and tasks, which might not be directly related to all real-world scenarios or practical applications.
2. **Lack of Guidance for Practice**: Beyond the insight that deeper and wider transformers have greater capacity, what guidance can this paper provide to practitioners?
3. **Some Impractical Settings**: The settings of blank tokens appended to the input sequence are not related to the practical scenarios.
4. **Need for Further Empirical Evaluation**: The experimental results of this paper are only performed on the GraphQA benchmark. More experimental results are needed to validate the theoretical results on real-world tasks.

**Questions:**

See the Weaknesses section.

**Limitations:**

Authors write about limitations fairly.

---

> ### Author Rebuttal · Authors · 2024-08-06
>
> We thank the reviewer for their comments and attention to the utility of the theoretical analysis.
>
> > Limited Generalizability: The theoretical results are based on specific assumptions, parameter scaling regimes, and tasks, which might not be directly related to all real-world scenarios or practical applications.
>
> While it is the nature of theoretical works to be reliant on specific (often artificial) regimes, we did our best effort to define concise tasks and use transformer formalisms that resemble modern models, while providing a foundation for subsequent work to build on.
>
> In particular, the parameter scaling assumptions employed were chosen to be consistent with their counterparts in both publicly available SOTA networks and theoretical models. Our principal scaling assumption---that the context length $N$ is much larger than the embedding dimension $m$ (e.g. $m = N^{0.1}$) , which is in turn much larger than the number of heads $H$ and depth $L$ (e.g. $L = \log N$)---is validated by models like Llama 3 405B (where $H = 128$, $L = 126$, $m = 16,384$, and $N = 128,000$). Furthermore, comparable theoretical results [3,4,5] all consider regimes where the context length is the primary quantity of interest and all other transformer parameters grow at a strictly lower rate than that.
>
> We discuss the arbitrary MLP assumption in our response to Reviewer CKNh, which may also be relevant.
>
> > Lack of Guidance for Practice: Beyond the insight that deeper and wider transformers have greater capacity, what guidance can this paper provide to practitioners?
>
> The primary goal of this paper is to understand empirically and theoretically the complexity of different tasks. While we don’t give prescriptive insights for training transformers, we hope that the framework introduced includes intuitions that help practitioners understand which tasks (graphical, linguistic, or otherwise) that transformers can be expected to learn.
>
> For instance, we establish theoretically that transformers have immense parallel computational potential (which is evident in the efficient theoretical constructions and positive learnability results for tasks like graph connectivity), which may suggest to practitioners that with the correct style of hints, they can go beyond the step-by-step reasoning that is used for chain-of-thought reasoning. Tasks like graph connectivity may be transferable to NLP settings and suggest that certain types of multi-step reasoning or synthesis of information across passages should be learnable. In contrast, search-type problems (like shortest path) are evidently more difficult, and suggest that practitioners should expect language models to struggle to navigate through complex chains of reasoning where the individual “hops” are not self-evident.
>
> > Some Impractical Settings: The settings of blank tokens appended to the input sequence are not related to the practical scenarios.
>
> The study of blank tokens is an area of active research. Many recent papers–theoretical and empirical–have investigated the utility of including blank tokens as “scratch-pad” tokens [6] (where models are evaluated auto-regressively and can output intermediate tokens of their choosing) or “pause” tokens [7] (where the blank tokens provide no new information, but can extend the computational powers of the architecture). We believe it is worthwhile to point out the theoretical benefits of including a large number of these tokens. At the same time, we also investigate other scaling regimes (e.g. the LDW regime) where pause tokens are not included.
>
> > Need for Further Empirical Evaluation: The experimental results of this paper are only performed on the GraphQA benchmark. More experimental results are needed to validate the theoretical results on real-world tasks.
>
> We restrict our focus to the GraphQA tasks because it provides us with a concise setting that provides revealing separations between different classes of models and has a close connection to theoretical settings. We would be interested in seeing future works study other problems, but we think these kinds of synthetic tasks improve the clarity and completeness of our paper’s message. Furthermore, the tasks we study are fundamental and often occur as subroutines of other problems, such as shortest path for navigation.
>
> [3] Bingbin Liu, Jordan T. Ash, Surbhi Goel, Akshay Krishnamurthy, Cyril Zhang. Transformers Learn Shortcuts to Automata
>
> [4] Alberto Bietti, Vivien Cabannes, Diane Bouchacourt, Herve Jegou, Leon Bottou. Birth of a Transformer: A Memory Viewpoint.
>
> [5] Clayton Sanford, Daniel Hsu, Matus Telgarsky.Transformers, parallel computation, and logarithmic depth.
>
> [6] Maxwell Nye, Anders Johan Andreassen, Guy Gur-Ari, Henryk Michalewski, Jacob Austin, David Bieber, David Dohan, Aitor Lewkowycz, Maarten Bosma, David Luan, Charles Sutton, Augustus Odena. Show Your Work: Scratchpads for Intermediate Computation with Language Models.
>
> [7] Sachin Goyal, Ziwei Ji, Ankit Singh Rawat, Aditya Krishna Menon, Sanjiv Kumar, Vaishnavh Nagarajan. Think before you speak: Training Language Models With Pause Tokens.

---

### Official Review · Reviewer_RdpW · 2024-07-12

**Soundness:** 3
**Presentation:** 4
**Contribution:** 3
**Rating:** 7
**Confidence:** 4

**Summary:**

In this work, the authors propose a new representational hierarchy for standard transformers in terms of computing algorithms over graphs. To this end, the authors relate transformers to the massively parallel computing model (MPC), thus establishing the connection to several graph algorithms. The empirical study validates the theoretical results. In particular, the authors compare transformers to graph neural networks (GNNs), as well as training small transformers from scratch, fine-tuning large-scale pre-trained language models and LLM prompting.

**Strengths:**

- The motivation for this work is strong and the study is timely. In particular, studying the representational capacity of transformers on graphs is important and useful both for studying the capabilities of transformers in general as well as for tasks explicitly modeling data as graphs.
- The paper is written clearly and concise. In particular, I appreciated the “Theoretical interpretation” parts below each empirical result which helps navigating the empirical study in the context of the theoretical results.
- The empirical study is tightly linked to the theory. For example, the presented theoretical results have implications on the tasks in GraphQA, the benchmark used in the empirical study and the empirical study backlinks to the corresponding theoretical result.

**Weaknesses:**

- There is an ambiguity in the empirical results in 4.1 and 4.2. I searched the experimental details in the main paper as well as Appendix E.3 and believe that the authors use standard GNN baselines without added positional encodings (also see my question to confirm this). However, note that the authors use transformers with a tokenization following TokenGT (https://arxiv.org/abs/2207.02505), which leverages positional encodings (in particular, node identifiers). Now, the authors for example say in L290-291 that GNNs do not have the sufficient representational capacity to compute graph connectivity. At the same time, the authors attribute the positive results in 4.2. to the inductive bias of GNNs (e.g., L319). I think these two claims should be distinguished. GNNs with node identifiers are universal (see e.g., Abboud et al. 2020, https://arxiv.org/abs/2010.01179) but arguably still possess the inductive bias mentioned in L319, namely that they exclusively aggregate information via the 1-hop neighborhood. Further, it is known in the graph learning community that GNNs with added positional encodings perform much stronger on long-range tasks; see Tönshoff et al. 2023, https://arxiv.org/abs/2309.00367). I suggest to compare to both GNNs with node identifiers as well as without and to clearly distinguish claims about expressivity from those of inductive bias.
- The authors state that standard deviation is not provided due to high runtime costs. However, I would appreciate if standard deviation could be provided at least for the smaller models, that is, the GNN baselines as well as the 60M transformer to obtain an understanding of the variance of the presented results.

**Questions:**

- Do I understand correctly that the graph data is provided to the LM-based models  as text, whereas the 60M transformer and the GNN baselines receive the graph information in terms of a custom tokenization (in the case of the transformer) or encoding (in the case of the GNN)?
- Do I understand correctly that the GNNs are not trained with node identifiers but the 60M transformer is (following TokenGT)?

**Limitations:**

The limitations were adequately addressed.

---

> ### Author Rebuttal · Authors · 2024-08-06
>
> We appreciate the reviewer’s attention to detail and questions about our experimental assumptions and rigor.
>
> >There is an ambiguity in the empirical results in 4.1 and 4.2. I searched the experimental details in the main paper as well as Appendix E.3 and believe that the authors use standard GNN baselines without added positional encodings (also see my question to confirm this). However, note that the authors use transformers with a tokenization following TokenGT (https://arxiv.org/abs/2207.02505), which leverages positional encodings (in particular, node identifiers). Now, the authors for example say in L290-291 that GNNs do not have the sufficient representational capacity to compute graph connectivity. At the same time, the authors attribute the positive results in 4.2. to the inductive bias of GNNs (e.g., L319). I think these two claims should be distinguished. GNNs with node identifiers are universal (see e.g., Abboud et al. 2020, https://arxiv.org/abs/2010.01179) but arguably still possess the inductive bias mentioned in L319, namely that they exclusively aggregate information via the 1-hop neighborhood.
>
> We thank the reviewer for pointing out this apparent contradiction. Our previous statement on L290-291 was unclear, and we propose to amend it to state the following:
>
> "In contrast, message-passing GNNs are unable to solve connectivity in a similarly **depth- and width-efficient** manner due to fundamental capacity limitations."
>
> We agree that MPNNs with node identifies are able to solve graph connectivity (owing to GNN universality), but the proofs of [2] (which we discuss in Appendix D) specify the impossibility of solving connectivity with GNNs of small-polynomial width and depth. The universality and inductive bias results, therefore, do not present a contradiction with the representational hardness results in the bounded-width regime. Taken together, these results suggest that favorable inductive biases of small GNNs will be unable to surmount the representational barrier presented in [2].
>
> > Further, it is known in the graph learning community that GNNs with added positional encodings perform much stronger on long-range tasks; see Tönshoff et al. 2023, https://arxiv.org/abs/2309.00367). I suggest to compare to both GNNs with node identifiers as well as without and to clearly distinguish claims about expressivity from those of inductive bias.
>
> We conducted further experimentation to validate that the inclusion of node identifiers did little to improve performance on a subset of the tasks. We can augment Figures 3a and Table 2 with additional rows reflecting different GNNs with node identifiers:
>
> | Model | Connectivity (1k samples) | Connectivity (100k) | Cycle Check (1k) | Cycle Check (100k) | Node Degree (1k) | Node Degree (100k) |
> | --- |  --- | --- | --- | --- | --- | --- |
> | GCN with identity tokenization | 83.8 | 93.2 | 83.2 | 91.2 | 8.8 | 42.2 |
> | MPNN with identity tokenization | 94.0 | 93.8 | 98.6 | 99.8 | 38.8 | 100.0 |
> | GIN with identity tokenization | 93.4 | 94.0 | 97.6 | 98.6 | 37.8 | 100.0 |
>
> Notably, in the case of connectivity, these represent very similar performance to their counterparts without node identifiers (with the exception of the GCN with 100k samples), and none come close to attaining the 98.0% accuracy for a 60M transformers with 100k samples. We intend to add these plots and a discussion of them to future versions of the paper.
>
> We appreciate the citations shared with us regarding GNN universality and positional encodings and will include them in a future version of the paper.
>
> > The authors state that standard deviation is not provided due to high runtime costs. However, I would appreciate if standard deviation could be provided at least for the smaller models, that is, the GNN baselines as well as the 60M transformer to obtain an understanding of the variance of the presented results.
>
> Thank you for your suggestion. We have conducted additional experiments to provide standard deviation for GIN and Transformer 60M models with 1k samples. After training five times with different random seeds, we observed an average accuracy of 93.64 with a standard deviation of 0.385 for GIN and an average accuracy of 92.705 with a standard deviation of 0.483 for the transformer 60M model. These closely resemble our singleton experiments in the submission. We are prepared to extend this analysis to other models if the reviewer believes it would enhance the paper.
>
> > Do I understand correctly that the graph data is provided to the LM-based models as text, whereas the 60M transformer and the GNN baselines receive the graph information in terms of a custom tokenization (in the case of the transformer) or encoding (in the case of the GNN)?
>
> > Do I understand correctly that the GNNs are not trained with node identifiers but the 60M transformer is (following TokenGT)?
>
> Yes, both are correct.
>
> [2] Andreas Loukas. What graph neural networks cannot learn: depth vs width.

---

> > ### Comment · Reviewer_RdpW · 2024-08-07
> >
> > I thank the authors for their rebuttal and for the additional empirical results. Regarding standard deviation, I suggest to repeat experiments for multiple random seeds where ever feasible. Other than that, my concerns are addressed.

---

### Official Review · Reviewer_CKNh · 2024-07-14

**Soundness:** 3
**Presentation:** 3
**Contribution:** 3
**Rating:** 7
**Confidence:** 4

**Summary:**

This paper studies the reasoning capabilities of transformers by characterizing their representational power to execute graph algorithms. Theoretically, the authors employ a general transformer model previously studied by Sanford et al. (2024), which assumes that the local MLPs can represent arbitrary functions and thus may have unbounded parameters. Using this transformer model, the authors investigate depth, width, and number of extra tokens required for algorithm execution. This leads to a representational hierarchy of graph reasoning tasks. In particular, by relating the transformer model to the MPC distributed computing model, the authors show that logarithmic depth is both sufficient and necessary to solve parallelizable graph tasks. Empirically, the authors carry out well-designed experiments to demonstrate their theoretical results.

**Strengths:**

- Understanding the reasoning capabilities of transformers is a highly important problem, both in theory and in practice. The results in this paper add new insights from the prospective of algorithm execution.
- The technical result on MPC simulation in this paper is a notable improvement from the previous result in Sanford et al. (2024). It is interesting to see that using extra tokens can significantly reduce the required embedding dimension.
- The experiments are well designed. The paper is well written. Overall I think this is a good paper with clear contributions.

**Weaknesses:**

- As the authors mentioned at the end of the paper, the assumption of unbounded-size MLPs provides strong results. At the same time, this is an apparent gap between theory and practice. I think that making such assumption is fine. Just raising this point as a limitation of the current theoretical framework.
- The graphs used in the experiments are super small (5 to 20 nodes) compared to the size of transformers (60M and 11B). Since, theoretically, the required depth and width are sublinear with respect to the input size, this empirical setting creates a huge gap between theory and practice. It would be nice if the authors carry out additional experiments on larger graphs, or limit the number of parameters in the transformers to a few thousands.

**Questions:**

- Have you tried experiments on much larger graphs? There is a huge gap in the parameter count of transformers and the size of graphs used in the current experiments.
- A related work [1] studies execution of graph algorithms using a looped transformer architecture. There, the authors prove that a transformer layer with constant depth and width can simulate a single algorithmic step of a graph algorithm for any input graph size. Consequently, by looping the same layer repeatedly, the resulting transformer model (having constant parameter count) can execute several graph algorithms. The authors should discuss the difference between their theoretical framework and that considered in [1]. Because it seems that the looping mechanism might help the transformer model to gain more efficiency in terms of parameter count.

[1] Simulation of Graph Algorithms with Looped Transformers. Artur Back de Luca, Kimon Fountoulakis. ICML 2024.

**Limitations:**

The authors should explicitly discuss about the limitations due to the assumption on local MLPs can represent arbitrary functions. For example, would this be a potential reason for the huge gap between the input sequence length (size of graph) and the transformer size (parameter count) in the experiments? The authors should comment on how likely will the assumption lead to a theory-practice gap, and possible future avenues to close this gap.

---

> ### Author Rebuttal · Authors · 2024-08-06
>
> We thank the reviewer for their detailed responses and their close attention to our assumptions.
>
> > As the authors mentioned at the end of the paper, the assumption of unbounded-size MLPs provides strong results. At the same time, this is an apparent gap between theory and practice. I think that making such assumption is fine. Just raising this point as a limitation of the current theoretical framework.
>
> > The authors should explicitly discuss about the limitations due to the assumption on local MLPs can represent arbitrary functions. For example, would this be a potential reason for the huge gap between the input sequence length (size of graph) and the transformer size (parameter count) in the experiments? The authors should comment on how likely will the assumption lead to a theory-practice gap, and possible future avenues to close this gap.
>
> We acknowledge the limitation of unbounded-size MLPs, and we are happy to include further discussion about this assumption.
>
> While the assumption when maximally exploited (e.g. solving NP-hard problems within each MLP) represents a significant theory-practice gap, we believe that the assumption is fairly well motivated for the kinds of problems that we discuss. The MPC protocols for tasks like graph connectivity do not involve solving hard computational problems in each machine (see e.g. [8]), which means the resulting transformer MLPs could be represented by a fairly simple circuit. This in turn translates to a ReLU network of small size.
>
> We think the theory-practice gap can be ameliorated by future theoretical work that models an MLP as a circuit with $o(N)$ gates and examines whether the results investigated here are possible in that regime. This would maintain theoretical elegance while ruling out computational exploitation of the current assumption. Because tasks like graph connectivity can be solved with relatively simple MLPs, we suspect that this is not the root cause of needing larger models for learnability. However, we’d be interested in exploring the trade-offs of MLP and self-attention parameter sizes in future work.
>
> We also think that it’s a fairly reasonable assumption because MLP parameters regularly make up a large fraction of total trainable parameters in modern transformers and thus reflect significant representational capability. Furthermore, it is important to note that this limitation of the positive results conversely improves the generality of the negative results.
>
> We propose adding a brief discussion on the previous topics. Would this satisfy the reviewer’s concerns?
>
> > The graphs used in the experiments are super small (5 to 20 nodes) compared to the size of transformers (60M and 11B). Since, theoretically, the required depth and width are sublinear with respect to the input size, this empirical setting creates a huge gap between theory and practice. It would be nice if the authors carry out additional experiments on larger graphs, or limit the number of parameters in the transformers to a few thousands.
>
> > Have you tried experiments on much larger graphs? There is a huge gap in the parameter count of transformers and the size of graphs used in the current experiments.
>
> We appreciate the question. In the context of this paper, our principal goal was to compare the relative difficulty of solving different tasks with different models. With its pre-existing LLM benchmarks and wide range of tasks, the GraphQA dataset was a particularly good fit for that study. While the graphs in GraphQA have a small number of vertices, they are frequently dense graphs, and often require several hundred tokens to be passed as input to specify each graph. Much larger graphs would be more difficult to tokenize in the current framework, and future work may need to explore other forms of graph encoding. We would like to see further work that examines how transformer capabilities scale as the size of the graphs increase, but that will require a different dataset and likely different approaches to sampling random graphs.
>
> We acknowledge that our transformers often have many more parameters than the sizes of the graphs, and that our theoretical results suggest that this need not be the case *representationally*. However, over-parameterization is likely necessary from a learnability perspective. While the theoretical results clearly state that small models suffice for graph connectivity, more parameters may be necessary to obtain useful gradients.
>
> > A related work [1] studies execution of graph algorithms using a looped transformer architecture. There, the authors prove that a transformer layer with constant depth and width can simulate a single algorithmic step of a graph algorithm for any input graph size. Consequently, by looping the same layer repeatedly, the resulting transformer model (having constant parameter count) can execute several graph algorithms. The authors should discuss the difference between their theoretical framework and that considered in [1]. Because it seems that the looping mechanism might help the transformer model to gain more efficiency in terms of parameter count.
>
> We recently became aware of this work and plan on discussing this in our related work section. It provides an excellent point of comparison to our work. From the parameter-count perspective, we agree that this regime offers a parameter-efficient model for solving tasks like shortest path. However, their approach does so by simulating procedures like Dijkstra’s algorithms step-by-step, which means the effective “depth” of the resulting models would be polynomial in $N$ and critically avoids taking advantage of the parallel processing capabilities of transformers. In a practical sense, we suspect that it would be very difficult to train a looped transformer model given the large number of required loops.
>
> [8] Alexandr Andoni, Clifford Stein, Zhao Song, Zhengyu Wang, Peilin Zhong. Parallel Graph Connectivity in Log Diameter Rounds.

---

> > ### Comment · Reviewer_CKNh · 2024-08-13
> >
> > I'd like to thank the authors for their detailed responses. My questions have been addressed. I'm more convinced that this is a good paper with clear contributions. I will increase my confidence score and maintain the current supportive score (7).

---

### Author Rebuttal · Authors · 2024-08-06

We thank the reviewers for their close reading, detailed feedback, and recognition of the value of this work. Their questions and critique help clarify and improve the messages of this paper.

While we respond to each author’s comments in their corresponding rebuttal, we would to highlight in particular the fact that we ran several new experiments based on the feedback we received from Reviewer RdpW:
- We adapted our GNN experiments to have identity node embeddings and contrasted those results with featureless GNNs in response to a question about GNN identifiers.
- We reran some of our 60M transformer and GIN experiments with multiple random seeds in response to a question about standard deviation.

We appreciate all of the work that went into the review process, and we look forward to further discussion.

---

### Decision · Program_Chairs · 2024-09-25

**Decision:**

Accept (poster)

**Comment:**

This paper studies the reasoning capabilities of transformers by examining their expressiveness in graph algorithms. Theoretically, the authors investigate the depth, width, and number of extra tokens required for algorithm execution and give a representational hierarchy of graph reasoning tasks. In particular, the authors demonstrate that logarithmic depth is both sufficient and necessary to solve parallelizable graph tasks. Moreover, they conduct some experiments to verify their theoretical findings.

The reviewers found the paper to be technically sound and impactful, with a recommendation for acceptance after addressing some weaknesses and concerns. This work will contribute to the understanding of transformer reasoning capabilities over graph structure.